# A RuBisCO-mediated carbon metabolic pathway in methanogenic archaea

Takunari Kono[1,2], Sandhya Mehrotra[3], Chikako Endo[1], Natsuko Kizu[4], Mami Matusda[5], Hiroyuki Kimura[6,7], Eiichi Mizohata[4], Tsuyoshi Inoue[4], Tomohisa Hasunuma[5], Akiho Yokota[1], Hiroyoshi Matsumura[8,*] & Hiroki Ashida[2,*]

Two enzymes are considered to be unique to the photosynthetic Calvin–Benson cycle: ribulose-1,5-bisphosphate carboxylase/oxygenase (RuBisCO), responsible for $CO_2$ fixation, and phosphoribulokinase (PRK). Some archaea possess *bona fide* RuBisCOs, despite not being photosynthetic organisms, but are thought to lack PRK. Here we demonstrate the existence in methanogenic archaea of a carbon metabolic pathway involving RuBisCO and PRK, which we term 'reductive hexulose-phosphate' (RHP) pathway. These archaea possess both RuBisCO and a catalytically active PRK whose crystal structure resembles that of photosynthetic bacterial PRK. Capillary electrophoresis-mass spectrometric analysis of metabolites reveals that the RHP pathway, which differs from the Calvin–Benson cycle only in a few steps, is active *in vivo*. Our work highlights evolutionary and functional links between RuBisCO-mediated carbon metabolic pathways in methanogenic archaea and photosynthetic organisms. Whether the RHP pathway allows for autotrophy (that is, growth exclusively with $CO_2$ as carbon source) remains unknown.

[1] Graduate School of Biological Sciences, Nara Institute of Science and Technology, 8916-5 Takayama-cho, Ikoma, Nara 630-0192, Japan. [2] Graduate School of Human Development and Environment, Kobe University, 3-11 Tsurukabuto, Nada-Ku, Kobe 657-8501, Japan. [3] Department of Biological Sciences, Birla Institute of Technology and Science, Pilani, Rajasthan 333031, India. [4] Department of Applied Chemistry, Graduate School of Engineering, Osaka University, 2-1 Yamadaoka, Suita, Osaka 565-0871, Japan. [5] Department of Chemical Science and Engineering, Kobe University, 1-1 Rokkodai, Nada-ku, Kobe 657-8501, Japan. [6] Department of Geosciences, Faculty of Science, Shizuoka University, Shizuoka 422-8529, Japan. [7] Research Institute of Green Science and Technology, Shizuoka University, Shizuoka 422-8529, Japan. [8] Department of Biotechnology, College of Life Sciences, Ritsumeikan University, 1-1-1 Noji-higashi, Kusatsu, Shiga 525-8577, Japan. * These authors jointly supervised this work. Correspondence and requests for materials should be addressed to H.M. (email: h-matsu@fc.ritsumei.ac.jp) or to H.A. (email: hiroki_ashida@people.kobe-u.ac.jp).

To date, six mechanisms have been identified for carbon fixation[1]. Of these, members of Archaea are believed to be able to use three, namely dicarboxylate-hydroxybutyrate cycle, hydroxypropionate-hydroxybutyrate cycle, and reductive acetyl-CoA pathway. The reductive acetyl-CoA pathway has been discussed as a model of the primordial $CO_2$-fixing mechanism[1]. The Calvin–Benson cycle is the predominant photosynthetic $CO_2$-fixing pathway[2]; however, it is considered to be an evolutionary late innovation and has not been reported in Archaea[1]. The Calvin–Benson cycle consists of three phases: carbon fixation, carbon reduction, and ribulose-1,5-bisphosphate (RuBP) regeneration. In this cycle, ribulose-1,5-bisphosphate carboxylase/oxygenase (RuBisCO) fixes $CO_2$ with RuBP to yield 3-phosphoglycerate (3-PGA), from which RuBP is regenerated[3]. Phosphoribulokinase (PRK) synthesizes RuBP from ribulose-5-phosphate (Ru5P) in the final step in RuBP regeneration[3,4]. RuBisCO and PRK are representative and unique enzymes of the photosynthetic Calvin–Benson cycle[5]. Interestingly, approximately half of the members of archaea that have currently been sequenced have a conserved gene for *bona fide* RuBisCO. Nevertheless, *PRK* genes and PRK activity have not been detected in Archaea. An alternative pathway distinct from the Calvin–Benson cycle may thus be present in Archaea to synthesize RuBP from either 5-phospho-D-ribose-1-pyrophosphate (PRPP) or nucleoside 5′-monophosphate (NMP) via common ribose-1,5-bisphosphate[6,7]. These pathways are not cyclic and cannot regenerate RuBP. The pentose bisphosphate pathway to generate RuBP as substrate for RuBisCO from nucleosides or NMP has been especially well characterized in *Thermococcus kodakarensis* and is thought to be involved in nucleoside assimilation and degradation[7–9].

Here, we report a carbon metabolic pathway involving RuBisCO and PRK that can regenerate RuBP in methanogenic archaea.

## Results

**Enzymatic analysis of archaeal PRK homologues.** Some members of the archaeal phylum Euryarchaeota, including *Methanospirillum hungatei*, possess conserved genes for PRK homologues, as well as RuBisCO. BLAST search with PRK of cyanobacterium *Synechococcus* PCC7942 against archaeal protein database identified PRK homologues in Archaea. Photosynthetic PRKs are classified into two groups from photosynthetic bacteria and plant-type oxygenic phototrophs such as plants, algae and cyanobacteria[4,10,11]. Archaeal PRK homologues show approximately 30% amino acid sequence identity with PRKs in photosynthetic organisms and form a clade clearly distinct from those of bacterial and plant-type PRKs in a phylogenetic tree (Fig. 1). PRK homologue genes are conserved in 11 mesophilic methanogenic archaea (Fig. 1; Supplementary Table 1). In addition to methanogenic archaea, four hyperthermophilic archaea species, including *Archaeoglobus profundus* placed near the base of a 16S rRNA-derived phylogenetic tree[12], possess PRK homologue genes. In contrast, these genes are not conserved in *Methanopyrus* spp. and *Methanocaldococcus jannaschii*, both hyperthermophilic methanogenic archaea species.

Archaeal PRK homologues possess conserved Walker A (P-loop) and B motifs, which are usually involved in binding of the phosphate groups of nucleoside triphosphate and coordination of a $Mg^{2+}$ ion, respectively (Supplementary Fig. 1). These motifs are strongly conserved in P-loop kinases that recognize various substrates as phosphorylation acceptors[13]. We analysed the specificity of the *M. hungatei* PRK (MhPRK) homologue for the known P-loop kinase substrates pantothenate, thymidine, uridine, cytidine, ribosylnicotinamide, fructose 6-phosphate,

AMP, ribose 5-phosphate and Ru5P (refs 14–19), but found that it showed kinase activity only for Ru5P. In an electrospray tandem mass spectrometry analysis, the product of the reaction catalysed by MhPRK with Ru5P as the substrate was identified as RuBP. This identification was based on the product's peak $(M–H)^-$ at $m/z$ 309 and a comparison of its fragmentation pattern with that of authentic RuBP (Supplementary Fig. 2). MhPRK had a $V_{max}$ of $29.28 ± 1.70\ \mu mol\ min^{-1}\ mg\ protein^{-1}$, a $K_m(Ru5P)$ of $0.28 ± 0.05\ mM$ and a $K_m(ATP)$ of $20.7 ± 1.7\ \mu M$ (Table 1). PRK activity was also detected in *M. hungatei* cell extracts ($84.14 ± 8.32\ nmol\ min^{-1}\ mg\ protein^{-1}$). Archaeal PRK homologues from *Methanoculleus marisnigri*, *Methanosaeta thermophila*, *Methanosaeta concilii*, and *A. profundus* also showed significant PRK activities (Table 1). The $V_{max}$ values of the archaeal PRKs were lower than those of photosynthetic PRKs, but the archaeal PRKs showed higher affinities for ATP than did the photosynthetic PRKs (Table 1). MhPRK utilized broad phosphate donor substrates, such as ATP, GTP, CTP and UTP. Kinase activities with CTP, UTP and GTP were respectively 74.35%, 60.48% and 93.96% of that with ATP (Supplementary Fig. 3). On the other hand, photosynthetic PRK as a phosphate donor is relatively specific to ATP (refs 20,21).

**Crystal structure of *M. hungatei* PRK.** To structurally characterize the archaeal PRK, we determined the crystal structure of MhPRK at 2.5-Å resolution (PDB ID 5B3F). MhPRK forms a dimer with dimensions of $\sim 105 × 50 × 40\ Å^3$ (Supplementary Fig. 4a). The two protomers within the dimer are related by approximately two-fold noncrystallographic symmetry. Each protomer consists of an eight-stranded mixed β-sheet (β1–6 and β8–9) core surrounded by several α-helices (α1–α9) and β-strands (β1′, β2′ and β7; Fig. 2a,c). This protomer structure is similar to that of the only photosynthetic PRK whose structure has been reported to date[22], that of photosynthetic bacterial PRK from *Rhodobacter sphaeroides* (RsPRK; PDB ID 1A7J; r. m. s. ds of 2.1 Å over 164 topologically equivalent Cα positions of MhPRK and RsPRK) (Fig. 2a–d and Supplementary Fig. 5). This structural similarity of MhPRK and RsPRK is in spite of their low amino acid sequence identity (25%). Conversely, MhPRK (homodimer) and RsPRK (homooctamer) differ in their quaternary structures and manner of dimerization (Supplementary Fig. 4). In dimeric MhPRK, the central strands β7, which consolidates the formation of the dimer with the larger dimer interface ($1694.6\ Å^2$), and β8 form the dimer interface. In RsPRK, strands β5, 6 and 9 and α-helix 6 participate in the formation of the dimer interface, and α-helix 7 is involved in octamer formation[22]. These structures involved in dimer and octamer formation are placed in the *C*-terminal domain. The *N*-terminal domain (1–198) of MhPRK resembles that of RsPRK, whereas the *C*-terminal domain (199–319 for MhPRK) is relatively different (Supplementary Fig. 1). Therefore, a low similarity in the *C*-terminal domain between MhPRK and RsPRK is consistent with differences in the manner of dimerization and quaternary structure of the two, which may lead to different enzymatic properties such as allosteric regulation of RsPRK (refs 23,24), and not MhPRK by NADH and AMP.

Two sulphate ions are positioned at the active site of the *N*-terminal domain at the edge of the dimer in MhPRK (Fig. 2c; Supplementary Figs 4a and 6). One interacts with Arg59, Arg62, Tyr98 and His100 side chains, and the other interacts with the main-chain amides of residues 27–31 in the P-loop along with a water molecule that interacts with the side chains of Ser31 and Thr32 (Fig. 2e; Supplementary Fig. 7a,c). Except for Arg62, these residues are highly conserved in all PRKs; they are responsible for binding of the phosphate group on Ru5P or ATP (Supplementary

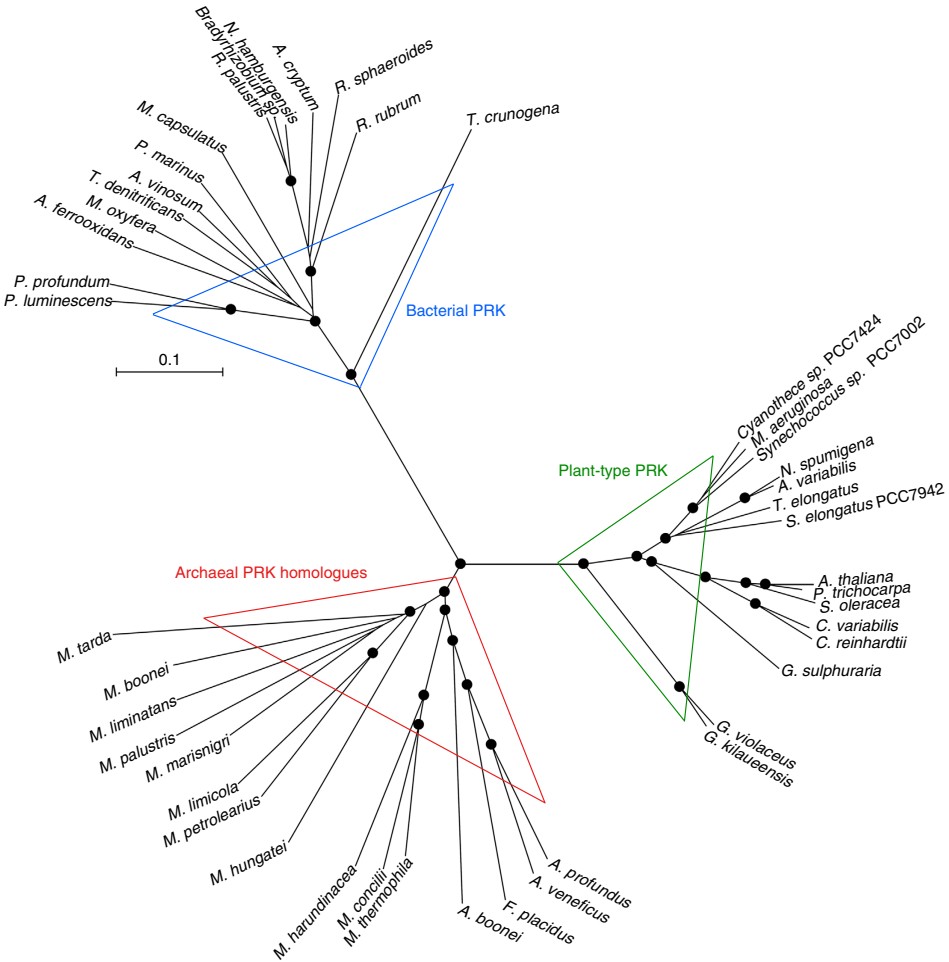

**Figure 1 | Phylogenetic tree of archaeal PRK and photosynthetic PRK.** The multiple sequence alignment and phylogenetic tree were produced using CLUSTALW. Bootstrap values were inferred from 1,000 replicates and significant bootstrapping values (>75%) are shown on the nodes as black filled circles. In eukaryotic PRKs, chloroplast transit peptides were removed according to ChloroP prediction[63]. PRKs are classified into three forms: plant-type from plants, algae and cyanobacteria (green triangle); bacterial-type from photosynthetic bacteria and chemoautotrophs (blue triangle); and archaeal-type (red triangles). Species abbreviations are as follows: *P. luminescens, Photorhabdus luminescens; P. profundum, Photobacterium profundum; A. ferrooxidans, Acidithiobacillus ferrooxidans; M. oxyfera, Methylomirabilis oxyfera; T. denitrificans, Thiobacillus denitrificans; A. vinosum, Allochromatium vinosum; P. marinus, Prochlorococcus marinus; M. capsulatus, Methylococcus capsulatus; R. palustris, Rhodopseudomonas palustris* BisA53; *N. hamburgensis, Nitrobacter hamburgensis; A. cryptum, Acidiphilium cryptum; R. sphaeroides, Rhodobacter sphaeroides; R. rubrum, Rhodospirillum rubrum; T. crunogena, Thiomicrospira crunogena; Cyanothece* sp. PCC 7424; *M. aeruginosa, Microcystis aeruginosa; Synechococcus* sp. PCC 7002; *A. variabilis, Anabaena variabilis; N. spumigena, Nodularia spumigena; T. elongatus, Thermosynechococcus elongatus; S. elongatus* PCC 7942, *Synechococcus elongatus* PCC 7942; *P. trichocarpa, Populus trichocarpa; A. thaliana, Arabidopsis thaliana; S. oleracea, Spinacia oleracea; C. variabilis, Chlorella variabilis; C. reinhardtii, Chlamydomonas reinhardtii; G. sulphuraria, Galdieria sulphuraria; G. violaceus, Gloeobacter violaceus; G. kilaueensis, Gloeobacter kilaueensis; M. marisnigri, Methanoculleus marisnigri; M. palustris, Methanosphaerula palustris; M. boonei, Methanoregula boonei; M. liminatans, Methanofollis liminatans; M. limicola, Methanoplanus limicola; M. petrolearius, Methanoplanus petrolearius; M. hungatei, Methanospirillum hungatei; M. tarda, Methanolinea tarda; M. concilii, Methanosaeta concilii; M. thermophile, Methanosaeta thermophila; M. harundinacea, Methanosaeta harundinacea; A. profundus, Archaeoglobus profundus; A. veneficus, Archaeoglobus veneficus; F. placidus, Ferroglobus placidus* and *A. boonei, Aciduliprofundum boonei.*

Fig. 1), as shown by the X-ray structure of the protein complexed with one sulphate ion in the binding site for the phosphate group on Ru5P, as well as mutagenesis studies of RsPRK (Fig. 2f; Supplementary Fig. 7b,d)[25–29]. Mutagenesis and structural studies on RsPRK have also identified other residues responsible for Ru5P binding (His45 and Lys165) and ATP binding (Arg168). These residues are also structurally and sequentially conserved in MhPRK (His55, Lys151 and Arg154; Fig. 2e,f; Supplementary Figs 1 and 7). These biochemical and structural analyses demonstrate that *M. hungatei* and other archaea possess PRKs.

***Bona fide* RuBisCO in *M. hungatei*.** RuBisCOs can be classified into three forms (Fig. 3): forms I and II are involved in the

Calvin–Benson cycle in photosynthetic organisms[30,31], while form III is associated with the pentose bisphosphate pathway/PRPP metabolism in members of the Archaea lacking PRK (ref. 7). *M. hungatei* RuBisCO was found to have a specific activity for the carboxylase reaction of $0.146 \pm 0.022$ $\mu$mol min$^{-1}$ mg protein$^{-1}$. *M. hungatei* thus has two key enzymes in the Calvin–Benson cycle. The RuBisCO of *M. hungatei* forms a new clade together with RuBisCOs from other methanogenic archaea harbouring PRK; this clade is distinct from the form-III RuBisCOs in the phylogenetic tree (Fig. 3). Archaeal form-III RuBisCOs should therefore be categorized into two groups: form III-a in mainly methanogenic archaea and form III-b in other archaea. Methanogenic archaea possessing form III-a RuBisCO also have PRK. The placement of

**Table 1 | Enzymatic parameters of PRKs from archaea and photosynthetic organisms.**

| Species | $V_{max}$ (μmol min$^{-1}$ mg protein$^{-1}$) | $K_m$(Ru5P) (mM) | $K_m$(ATP) (μM) |
|---|---|---|---|
| *Methanospirillum hungatei* (Archaeon) | 29.28 ± 1.70 | 0.28 ± 0.05 | 20.70 ± 1.70 |
| *Methanoculleus marisnigri* (Archaeon) | 36.77 ± 3.09* | N.A. | N.A. |
| *Methanosaeta thermophila* (Archaeon) | 19.10 ± 0.59 | 0.23 ± 0.05 | 5.66 ± 0.21 |
| *Methanosaeta concilii* (Archaeon) | 43.84 ± 2.84 | 0.31 ± 0.08 | 17.91 ± 2.02 |
| *Archaeoglobus profundus* (Archaeon) | 1.68 ± 0.06* | N.A. | N.A. |
| *Spinacia oleracea* (Plant) | 410[†] | 0.22[†] | 62[†] |
| *Synechococcus elongatus* PCC 7942 (Cyanobacterium) | 230[‡] | 0.27[‡] | 90[‡] |
| *Rhodobacter sphaeroides* (Bacterium) | 338[§] | 0.10[§] | 550[§] |
| *Halothiobacillus neapolitanus* (Bacterium) | 50[‖] | 0.24[‖] | 710[‖] |

Data for archaeal PRKs in this study are means ± s.d. of three replicates. N.A., not analysed.
*Specific activity.
[†]Porter et al.[65]
[‡]Kobayashi et al.[49]
[§]ATP concentration at half-maximal rate, as calculated from fitting ATP saturation data to the Hill equation, Runquist and Miziorko[25].
[‖]MacElroy et al.[66]

some methanogenic archaeal RuBisCOs in the form-II clade of photosynthetic bacteria reveals the evolutionary intersection between nonphotosynthetic and photosynthetic organisms with respect to RuBisCO. On the other hand, archaeal PRK forms a clade that is clearly separated from bacterial and plant-type clades (Fig. 1), which suggests that the molecular evolution of RuBisCO and PRK has been different.

**A carbon metabolic pathway involving RuBisCO.** The coexistence of PRK and RuBisCO indicated that $CO_2$ fixation might exist in *M. hungatei*. However, genes encoding three other Calvin–Benson cycle enzymes—transketolase, sedoheptulose-1,7-bisphosphatase, and ribulose-5-phosphate 3-epimerase—that catalyse the steps from fructose-6-phosphate (F6P) to Ru5P are missing from this archaeon, as they are from most members of the Archaea. In general, Archaea do not possess genes for transketolase and transaldolase needed for the non-oxidative pentose phosphate pathway[32,33]; instead, aromatic amino acids are synthesized via chorismate and shikimate in the 6-deoxy-5-ketofructose-1-phosphate pathway[34]. Archaea may synthesize Ru5P from F6P via the ribulose monophosphate (RuMP) pathway operating in the reverse direction[33,35]. In the RuMP pathway, formaldehyde is fixed with Ru5P to form D-arabino-3-hexulose-6-phosphate (Hu6P) by D-arabino-3-hexulose-6-phosphate synthase (HPS) and then isomerized to F6P by 6-phospho-3-hexuloisomerase (PHI)[36]. Both enzymes catalyse reversible reactions. *M. hungatei* has four candidate genes for the RuMP pathway. Two candidate HPSs are fusion proteins: *Mhun0647* (HPS-MenG) is a fusion of HPS and a domain similar to demethylmenaquinone methyltransferase (MenG)[37], whereas *Mhun1628* (Fae-HPS) is a fusion of HPS and formaldehyde-activating enzyme (Fae)[35], which catalyses the condensation of formaldehyde with tetrahydromethanopterin. MenG catalyses the last step in menaquinone biosynthesis, but the function of the MenG-like domain in HPS-MenG is unknown. HPS-MenG and Fae-HPS showed specific activities of 0.21 ± 0.05 and 6.05 ± 1.64 μmol min$^{-1}$ mg protein$^{-1}$, respectively (Supplementary Table 2). Two putative PHIs, *Mhun0910* and *Mhun3031*, named PHI-a and PHI-b in this report, jointly showed PHI activity (0.71 ± 0.17 μmol min$^{-1}$ mg protein$^{-1}$). However, there was no activity with PHI-a or -b alone (Supplementary Table 2), suggesting that PHI-a and -b catalyse the isomerase reaction in concert. We therefore concluded that these reaction steps were catalysed at biologically reasonable rates by HPS, PHI-a and PHI-b.

These results suggest that RuBP is regenerated from F6P via Hu6P by PHI-a and PHI-b, HPS-MenG/Fae-HPS and PRK, to serve as the substrate for RuBisCO in *M. hungatei*. This pathway might be cyclized by gluconeogenesis enzymes. Therefore, we propose that a carbon metabolic pathway involving RuBisCO and PRK exists in some methanogenic archaea, similar to the photosynthetic Calvin–Benson cycle. We term this pathway the 'reductive hexulose-phosphate' (RHP) pathway (Fig. 4). In extracts of *M. hungatei*, 3-PGA was produced from RuBP (Fig. 5a). In contrast, no 3-PGA was produced in the presence of 2-carboxy-D-arabinitol-1,5-bisphosphate (CABP), a RuBisCO-specific transition state inhibitor[38]. The fact that CABP completely inhibited the conversion of F6P into 3-PGA in *M. hungatei* extracts (Fig. 5b) shows that this conversion is RuBisCO-dependent.

To analyse the RHP pathway *in vivo*, a metabolomic analysis was performed with $^{13}C$-labelling from $NaH^{13}CO_3$ and *M. hungatei* living cells under heterotrophic conditions. Capillary electrophoresis-mass spectrometry (CE–MS) was used to analyse the $^{13}C$-labelling rate of compounds. Intracellular metabolites were extracted and analysed by CE–MS after $^{13}C$ labelling for 1, 3, 5 and 10 min to access metabolic turnover. The ratio of $^{13}C$ to total carbon in each metabolite, the $^{13}C$-fraction (%), was calculated from mass isotopomer distributions. The $^{13}C$-fraction associated with sugar phosphates involved in the RHP pathway, gluconeogenesis and glycolysis increased linearly depending on time, with particularly high $^{13}C$-labelling rates observed for 3-PGA, fructose-1,6-bisphosphate (FBP), F6P, glucose-6-phosphate (G6P), 2-phosphoglycerate (2-PGA) and phosphoenolpyruvate (PEP) (Fig. 6; Supplementary Fig. 8). Considering that the RuBisCO was active in extracts and labelling rate and pool size of 3-PGA were high (Supplementary Fig. 9), 3-PGA was expected to be the first major compound with incorporated $^{13}CO_2$ in *M. hungatei*. On the other hand, $^{13}C$ atoms were incorporated into acetyl-CoA at a low rate, but hardly into pyruvate, which suggests that the reductive acetyl-CoA pathway worked slowly and that the carbon flow from acetyl-CoA to 3-PGA via pyruvate, PEP, and 2-PGA did not play a major role in $^{13}C$ atom incorporation into 3-PGA under our experimental conditions. These results support the idea that incorporation of $^{13}C$ atoms into 3-PGA mainly stemmed from the carboxylase reaction of RuBisCO. Among the intermediates of the RHP pathway, FBP and F6P showed high $^{13}C$ labelling rates and Ru5P showed a significant, albeit lower, one, strongly suggesting that this pathway was active in *M. hungatei*. The $^{13}C$ labelling was not detected in RuBP, glyceraldehyde-3-phosphate (GAP), and dihydroxyacetone phosphate (DHAP) because the original pool sizes of these

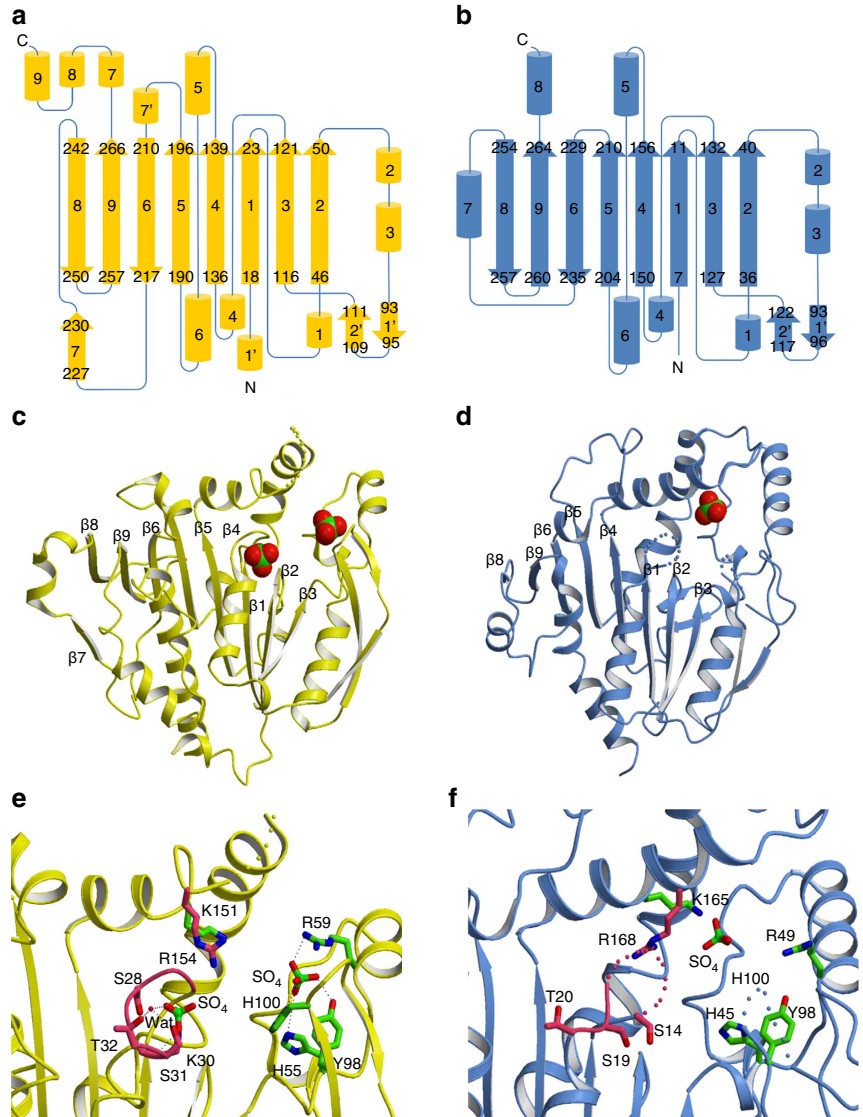

**Figure 2 | Structural comparison of archaeal and photosynthetic PRKs.** (**a,b**) Topology diagrams of the folding patterns in protomers of (**a**) *M. hungatei* PRK (MhPRK) and (**b**) *R. sphaeroides* PRK (RsPRK) are shown in yellow and blue, respectively. α-helices are denoted by cylinders, β-sheets by arrows and connecting loops by lines. Positions in the sequence that start and end each major secondary structural element are shown. (**c,d**) Ribbon diagrams of (**c**) MhPRK (Protein Data Bank (PDB) ID 5B3F) monomer and (**d**) RsPRK (PDB ID 1A7J) monomer. Sulphate ions are bound to the active site of MhPRK and RsPRK. Disordered regions of missing electron density are shown as dots for residues for 156–163 in MhPRK, residues 15–17 corresponding to a part of the P-loop, and residues 100–105 in RsPRK. (**e,f**) Active-site structures of (**e**) MhPRK and (**f**) RsPRK. Side chains of residues involved in ATP- and Ru5P-binding are shown as pink and green sticks, respectively, with oxygen atoms in red and nitrogen atoms in blue. Sulphate ions are shown in MhPRK and RsPRK, with sulphur atoms in green, and oxygen and nitrogen atoms in the same colours as those in residue side chains. Small red balls represent water molecules. Disordered regions are shown as dots, and P-loop containing residues involved in ATP binding are shown in pink. Dotted lines show interactions of active site sulphate ions.

unlabelled metabolites were very low (Supplementary Fig. 9). A large proportion of the $^{13}C$ atoms incorporated into 3-PGA were effluxed from the RHP pathway through F6P and 3-PGA to gluconeogenesis and glycolysis, respectively. The metabolic flow of glycolysis from 3-PGA to PEP via 2-PGA was clearly detected but that from PEP to acetyl-CoA via pyruvate was not, because time course of the $^{13}C$-fraction were largely different between PEP and pyruvate. The metabolic flow of gluconeogenesis, from F6P to G6P and G1P was observed. These results suggest that a large proportion of carbons fixed by RuBisCO were supplied to gluconeogenesis and glycolysis, and a small part of these were recycled for RuBP regeneration in the RHP pathway.

## Discussion

The results of our *in vitro* and *in vivo* experiments support the idea that RuBisCO and PRK function in the RHP pathway to fix $CO_2$. In addition, we found that this pathway was the major mode of $CO_2$ fixation in *M. hungatei* under our experimental heterotrophic conditions.

However, the $^{13}C$-atom labelling rate was much lower and the pool size of 3-PGA much smaller compared with observations in photosynthetic organisms[39,40]. This suggests that investment of energy in the RHP pathway is much smaller compared with that invested by plants and cyanobacteria in the Calvin–Benson cycle. Methanogenic archaea carry out methanogenesis with a low energy output compared with photosystems in photosynthesis,

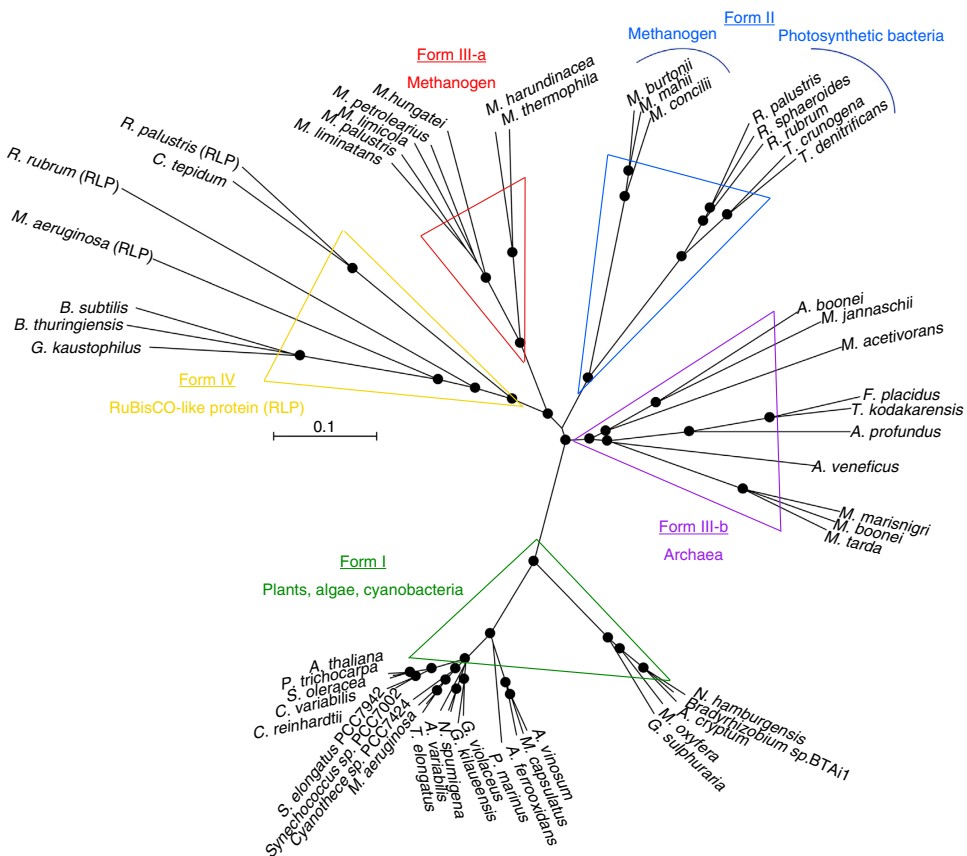

**Figure 3 | Phylogenetic tree of RuBisCOs and RuBisCO-like proteins.** The phylogenetic tree was produced using CLUSTALW. Bootstrap values were inferred from 1,000 replicates and significant bootstrapping values (>75%) are shown on the nodes as black filled circles. RuBisCO clades are indicated as follows: green for form I, blue for form II, red for form III-a and purple for form III-b (methanogenic archaea). We propose the latter two novel small clades because form III is prominently divided. The form IV clade of RuBisCO-like proteins (RLPs), which function as enolases/isomerases in methionine recycling in some bacteria, is shown in yellow[64]. Species abbreviations are as follows: *C. tepidum*, *Chlorobium tepidum*; *B. subtilis*, *Bacillus subtilis*; *G. kaustophilus*, *Geobacillus kaustophilus*; *B. thuringiensis*, *Bacillus thuringiensis*; *M. burtonii*, *Methanococcoides burtonii*; *M. mahii*, *Methanohalophilus mahii*; *M. jannaschii*, *Methanocaldococcus jannaschii*; *M. acetivorans*, *Methanosarcina acetivorans*; *T. kodakarensis*, *Thermococcus kodakarensis*; *R. palustris*, *Rhodopseudomonas palustris* and *N. spumigena*, *Nodularia spumigena*.

which is consistent with the observation that *M. hungatei* RuBisCO and PRK activities were much lower than those of photosynthetic organisms (Table 1). In the regeneration of RuBP during the RHP pathway, carbon is fixed as formaldehyde; one molecule of $CO_2$ is reduced to one molecule of formaldehyde using three molecules of ATP and two molecules of NAD(P)H (Fig. 4). Considering that methanogenic archaea thrive under extreme energy limitation, RuBP regeneration may incur a high energy cost for these species. The energy yield of methanogenesis under standard conditions is estimated to be 131 kJ mol$^{-1}$ of methane produced. Under ecological conditions of 1–10 Pa $H_2$, the energy yield is lower at 15–35 kJ mol$^{-1}$ and this is insufficient for the synthesis of 1 mol of ATP per 1 mol of methane produced[41]. Methanogenesis coupled with ATP synthesis depends on substrate $H_2$ concentrations[42]. Therefore, RuBP regeneration can probably work under conditions where methanogenesis is sufficiently active to produce enough energy for the RHP pathway and other pathways essential for growth.

Our experimental conditions included high hydrogen and $CO_2$ concentrations, which should sustain active methanogenesis. In addition, a heterotrophic medium containing formate, acetate, yeast extract and trypticase peptone was used for culture. Although *M. hungatei* can grow under autotrophic and heterotrophic conditions[43], a heterotrophic medium has been established for a suitable pure culture of *M. hungatei* in labo-

ratories and enables to grow with higher rate than autotrophic medium. These heterotrophic conditions might also provide energy for the RHP pathway. Whether the RHP pathway allows for autotrophy, that is, growth exclusively with $CO_2$ as carbon source, is unknown at present.

CE–MS analysis showed that a large proportion of the carbon fixed by RuBisCO flowed to gluconeogenesis and glycolysis, and that a small fraction of this carbon was released as formaldehyde in RuBP regeneration (Fig. 6). This suggests that the metabolic flux of RuBP regeneration in the RHP pathway was small and required relatively low energy, which can be sustained by energy obtained from methanogenesis. However, the energetics of the RHP pathway in methanogenic archaea under ecological conditions remains to be addressed further.

The RHP pathway supplies fixed carbon to other important metabolic pathways. CE–MS analysis showed that the RHP pathway provides a large proportion of fixed carbon to gluconeogenesis and glycolysis via F6P and 3-PGA. The RHP pathway also provides Ru5P, which is isomerized to ribose-5-phosphate before use, for nucleotide biosynthesis (Fig. 6). Production of Ru5P is likely to be completely dependent on the RHP pathway (Fig. 4) because *M. hungatei*, like most members of the Archaea, does not contain the non-oxidative pentose phosphate pathway. These effluxes of intermediates from the RHP pathway necessitate their replenishment

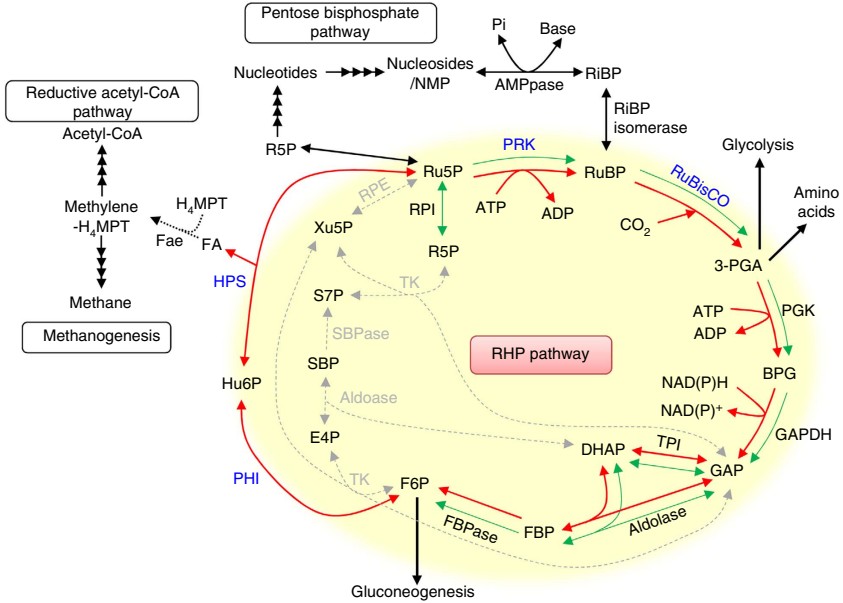

**Figure 4 | Proposed RHP pathway and related metabolic processes in Archaea.** The RHP pathway (highlighted in yellow) has metabolic links to methanogenesis, reductive acetyl-CoA pathway, pentose bisphosphate pathway, glycolysis, gluconeogenesis, and amino acid metabolism. Methanogenesis and reductive acetyl-CoA pathways share a metabolic intermediate, methylene-$H_4$MPT that might be synthesized by Fae with released formaldehyde from the RHP pathway (dotted black line). The successive black arrows show multiple reaction steps. The RHP pathway (red lines and arrows) is superimposed on the Calvin–Benson cycle (green lines and arrows), and reaction steps from Ru5P to F6P are common in both cycles. Missing Calvin–Benson-cycle steps in *M. hungatei* are indicated by grey dashed lines and arrows. Ru5P, ribulose-5-phosphate; RuBP, ribulose-1,5-bisphosphate; 3-PGA, 3-phosphoglycerate; BPG, 1,3-diphosphoglycerate; GAP, glyceraldehyde-3-phosphate; DHAP, dihydroxyacetone phosphate; FBP, fructose-1,6-bisphosphate; F6P, fructose-6-phosphate; Hu6P, D-arabino-3-hexulose-6-phosphate; FA, formaldehyde; E4P, erythrose-4-phosphate; Xu5P, xylulose-5-phosphate; SBP, sedoheptulose-1,7-bisphosphate; S7P, sedoheptulose-7-phosphate; R5P, ribose-5-phosphate; NMP, nucleoside 5'-monophosphate; RiBP, ribose-1,5-bisphosphate; $H_4$MPT, tetrahydromethanopterin; MF, methanofuran; PGK, 3-phosphoglycerate kinase; GAPDH, glyceraldehyde-3-phosphate dehydrogenase; FBPase, fructose-1,6-bisphosphatase; TK, transketolase; SBPase, sedoheptulose-1,7-bisphosphatase; RPE, ribulose-5-phosphate 3-epimerase; RPI, ribose-5-phosphate isomerase; AMPpase, AMP phosphorylase; RiBP isomerase, ribose-1,5-bisphosphate isomerase.

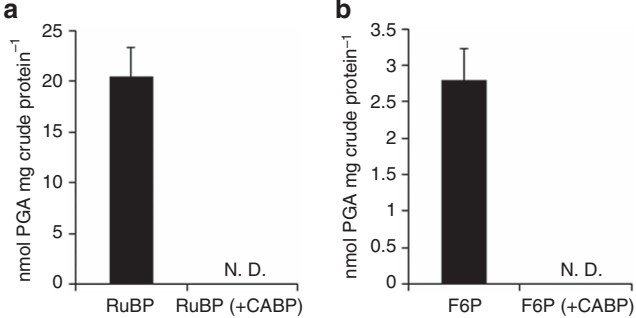

**Figure 5 | RuBisCO-dependent conversion of RuBP and F6P into 3-PGA in *M. hungatei* extracts.** Production of 3-PGA from RuBP (**a**) or F6P (**b**) by *M. hungatei* cell extracts, in the presence (open bars) and absence (black bars) of CABP, a RuBisCO-specific transition state inhibitor. CABP was added to the pre-activation mixture to inactivate RuBisCO. All assays were performed after reaction with the substrates for 2 h at 37 °C. Data are means ± s.d. of three replicates. N.D., not detected.

by anaplerotic reactions. One possibility is replenishment of 3-PGA from acetyl-CoA, and *M. hungatei* possesses genes for enzymes to catalyse metabolic steps from acetyl-CoA to 3-PGA, including pyruvate ferredoxin oxidoreductase (*Mhun0450*, γ subunit; *Mhun0451*, δ subunit; *Mhun0452*, α subunit; and *Mhun0453*, β subunit), phosphoenolpyruvate synthase (*Mhun2610*), phosphopyruvate hydratase (*Mhun1018*, *Mhun1101* and *Mhun2893*), and phosphoglycerate mutase (*Mhun0447* and *Mhun2324*). However, CE–MS data suggested

that this metabolic flow had little or no contribution to replenishment of 3-PGA. *M. hungatei* possesses conserved genes for the pentose bisphosphate pathway to synthesize RuBP from nucleosides or NMP (Supplementary Table 1), suggesting that RuBP may be provided by both the RHP pathway and the pentose bisphosphate pathway (Fig. 4). The heterotrophic medium used in our experiments provides nucleosides and NMP for the anaplerotic pentose bisphosphate pathway, to supply RuBP to the RHP pathway, as previously reported in *Thermococcus kodakaraensis*[9]. Hence, the pentose bisphosphate pathway may act as an anaplerotic pathway for the RHP pathway under our experimental conditions.

Formaldehyde released from the RHP pathway can be condensed by Fae with tetrahydromethanopterin to form methylene tetrahydromethanopterin, an intermediate in both the reductive acetyl-CoA pathway and methanogenesis for energy production, another pathway for $CO_2$ fixation (Fig. 4)[1,41,44]. The *M. hungatei* HPS is fused to Fae, which might immediately capture formaldehyde[35]. Archaea that have genes for the RHP pathway (except *Aciduliprofundum boonei*) either possess homologous genes for HPS and Fae (for example, in *Ferroglobus placidus*, *Methanosaeta harundinacea*, *M. concilii* and *M. marisnigri*) or have a gene that encodes a fused HSP/Fae protein (in other 10 species), similar to *M. hungatei* (Supplementary Table 1). Almost all archaea harbouring the RHP pathway possess a conserved Fae/HPS fusion protein (Supplementary Table 1). The fact that HPS and Fae are fused indicates that Fae might recover formaldehyde released from the RHP pathway and supply it to the reductive acetyl-CoA pathway and methanogenesis (Fig. 4). The fusion of HPS and Fae might

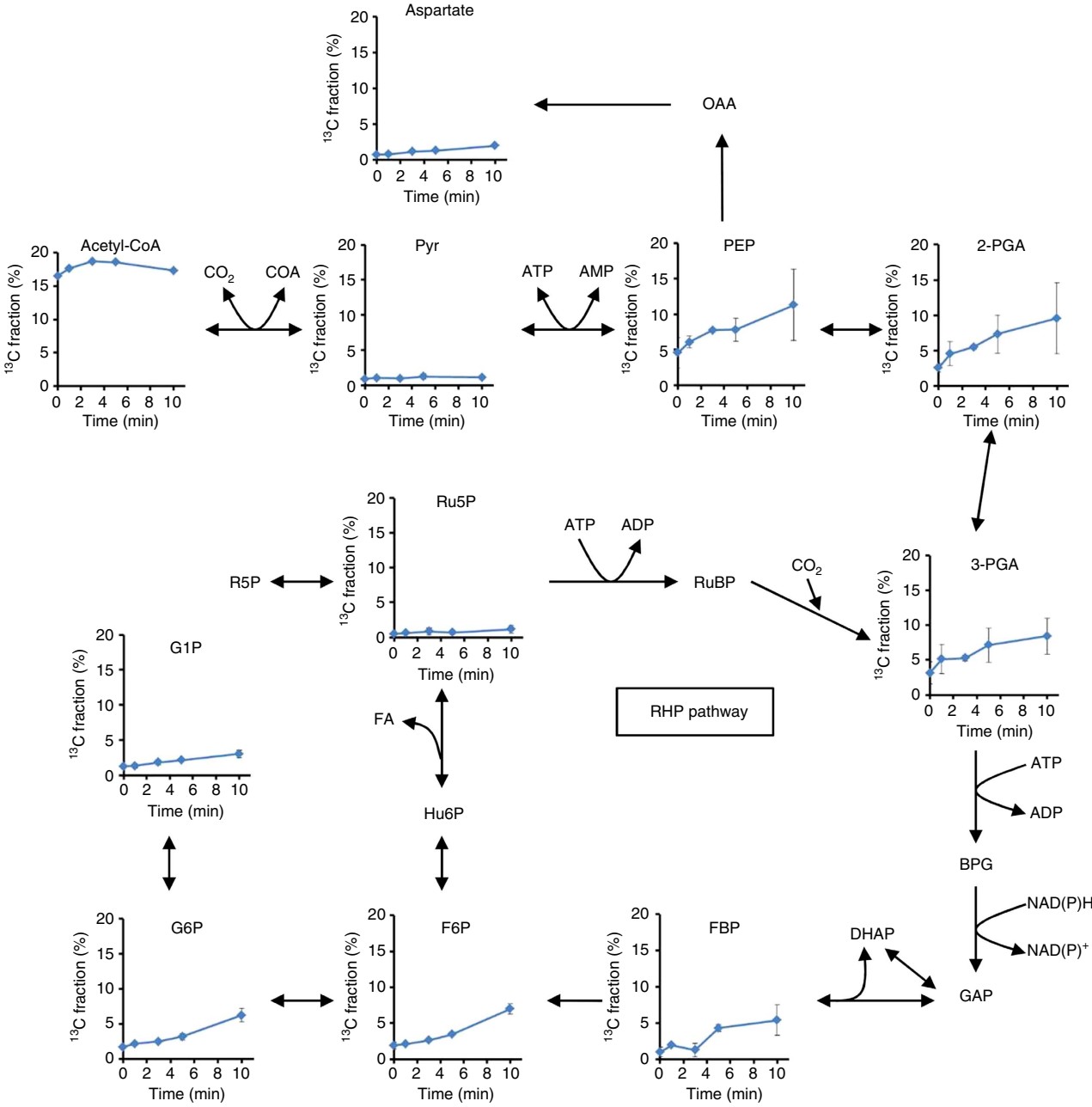

**Figure 6 | Time-course analysis of the metabolite $^{13}$C fraction of *M. hungatei* cells.** The $y$ axis represents the ratio of $^{13}$C to total carbon in each metabolite. Data are means ± s.d. of two replicates. Ru5P, ribulose-5-phosphate; RuBP, ribulose-1,5-bisphosphate; 3-PGA, 3-phosphoglycerate; BPG, 1,3-diphosphoglycerate; GAP, glyceraldehyde-3-phosphate; DHAP, dihydroxyacetone phosphate; FBP, fructose-1,6-bisphosphate; F6P, fructose-6-phosphate; Hu6P, D-arabino-3-hexulose-6-phosphate; FA, formaldehyde; R5P, ribose-5-phosphate; G6P, glucose-6-phosphate; G1P, glucose-1-phosphate; 2-PGA, 2-phosphoglycerate; PEP, phosphoenolpyruvate; Pyr, pyruvate; OAA, oxaloacetate.

minimize carbon loss and supply reduced carbon to the reductive acetyl-CoA pathway and methanogenesis. The CE–MS data showed that acetyl-CoA was not converted to 3-PGA via pyruvate ferredoxin oxidoreductase, phosphoenolpyruvate synthase, phosphopyruvate hydratase, and phosphoglycerate mutase (Fig. 6). Therefore, it is expected that formaldehyde is incorporated into acetyl-CoA in the reductive acetyl-CoA pathway and then converted to lipid via the mevalonate pathway, which is the archaeal isoprenoid biosynthetic pathway[45]. Nevertheless, the metabolomic analysis used in our study could not follow carbon from formaldehyde to methane

and acetyl-CoA. To clarify these issues, a metabolomic analysis system specific for metabolites produced by these pathways will need to be established.

Many methanogenic archaea contain conserved genes for enzymes in the RHP pathway, including RuBisCO and PRK, and for the reductive acetyl-CoA pathway (Supplementary Table 1). These pathways are thus expected to be widely distributed in methanogenic archaea. Hyperthermophilic archaea of the genera *Archaeoglobus* and *Ferroglobus* also contain genes for RHP pathway enzymes (Supplementary Table 1). These archaea also have conserved genes for methanogenesis; they can produce trace

amounts of methane[46] and have a reductive acetyl-CoA pathway[1]. This indicates that a metabolic link exists between the RHP pathway and the methanogenesis/reductive acetyl-CoA pathway.

Evolutionary acquisition of the photosynthetic Calvin–Benson cycle has had a major impact on the global carbon cycle[47]. The RHP pathway and the Calvin–Benson cycle only differ in a few steps, namely from F6P to Ru5P. Thus, our results shed light on evolutionary and functional links between carbon metabolic pathways involving RuBisCO in methanogenic archaea and photosynthetic organisms. We speculate that the photosynthetic Calvin–Benson cycle may have originated from a primitive carbon metabolic pathway utilizing RuBisCO, such as the archaeal RHP pathway, by replacement of some steps without release of carbon.

## Methods

**Multiple sequence alignment and phylogenetic analyses.** Amino acid sequences were aligned with Genetyx ver. 11, followed by manual adjustment. Phylogenetic analyses were conducted using the neighbour-joining (NJ) method with Genetyx ver. 11. Bootstrap values were inferred from 1,000 replicates. Trees were constructed using Genetyx-tree.

**M. hungatei growth conditions.** M. hungatei JF-1 ( $=$ JCM 10133$^T$ $=$ ATCC 27890$^T$ $=$ DSM 864$^T$ $=$ NBRC 100397$^T$) was obtained from the Japan Collection of Microorganisms (JCM), RIKEN BRC, a participant in the National BioResource Project of the Ministry of Education, Culture, Sports, Science and Technology, Japan. M. hungatei was cultivated under strictly anaerobic conditions at 37 °C for 2 weeks in JCM medium 242 (per litre: 0.3 g KH$_2$PO$_4$, 0.3 g K$_2$HPO$_4$, 0.3 g (NH$_4$)$_2$SO$_4$, 0.6 g NaCl, 0.13 g MgSO$_4 \cdot$7H$_2$O, 2.0 mg FeSO$_4 \cdot$7H$_2$O, 8.0 mg CaCl$_2 \cdot$2H$_2$O, 2.0 g yeast extract (Difco, BD, Japan), 2.0 g trypticase peptone (BBL, BD, Japan), 2.5 g sodium acetate, 2.5 g sodium formate, 10.0 ml trace mineral supplement (ATCC), 10 ml vitamin supplement (ATCC), 5.0 g NaHCO$_3$, 0.5 g Na$_2$S·9H$_2$O, 0.5 g L-cysteine·HCl·H$_2$O and 1.0 mg resazurin). The inoculated culture vessels were pressurized to 150 kPa with a H$_2$/CO$_2$ (80:20, v/v) gas mixture[48].

**Preparation of recombinant proteins.** Expression plasmids for the His$_6$-tagged M. hungatei PRK homologue and RuBisCO recombinant proteins were constructed as follows. Template genomic DNA was prepared from M. hungatei cells with a DNA plant mini kit (Qiagen, Japan). We used the Escherichia coli DH5α strain and the vector pET15b (Novagen: Merck, Japan) for cloning. A DNA fragment containing the gene encoding the PRK homologue (Mhun0794) with 5'-XhoI and 3'-NdeI sites, or RuBisCO (Mhun2315) with 5'-XhoI and 3'-BamHI sites, was amplified by polymerase chain reaction (PCR) with PrimeSTAR Max DNA polymerase (Takara, Japan) and the primer set Mhprk-F and R, or Mhrubisco-F and R (Supplementary Table 3), respectively, using M. hungatei genomic DNA as the template. After digestion with restriction enzymes, the fragments were inserted into pET-15b (Novagen) digested with the same restriction enzymes, resulting in the plasmids pET-Mhprk and pET-Mhrubisco. E. coli C43 (DE3) (Lucigen, WI, USA) was used to overexpress the M. hungatei PRK homologue and RuBisCO. Cells of E. coli were grown at 37 °C in LB medium containing 50 µg ml$^{-1}$ ampicillin until their optical density at 600 nm reached 0.4–0.6. Then, gene expression was induced by adding 1 mM β-isopropyl-D-thiogalactopyranoside and culturing the cells for 20 h at 18 °C. The cells were harvested by centrifugation (5,000g, 15 min).

Harvested cells were suspended in binding buffer (20 mM Tris–HCl, 500 mM NaCl, 5 mM imidazole, pH 8.0) and disrupted by sonication. The crude extracts were centrifuged at 20,000g for 15 min. The supernatant was applied to His-Bind resin (Novagen) equilibrated with binding buffer. The column was washed with binding buffer and wash buffer (20 mM Tris–HCl, 500 mM NaCl and 60 mM imidazole, pH 8.0), and then Mh-PRK and Mh-RuBisCO were eluted with elution buffer (20 mM Tris–HCl, 500 mM NaCl and 1 M imidazole, pH 8.0). Eluted samples were concentrated using an Amicon Ultra-10 K membrane concentrator (Merck Millipore, Japan). The buffer was exchanged with thrombin buffer (20 mM Tris–HCl, 150 mM NaCl, and 2.5 mM CaCl$_2$, pH 8.0) by using a PD-10 column (GE Healthcare, Japan). His-tagged Mh-PRK and Mh-RuBisCO were cleaved for 3 h at 25 °C with 2 units thrombin protease (Novagen). The His-tag and thrombin were removed by gel filtration on a Superose 6 10/300 GL column (GE Healthcare) eluted with gel filtration buffer (50 mM Tris–HCl, 0.5 mM EDTA and 100 mM KCl, pH 8.0).

To prepare the four other archaeal PRK homologues, expression plasmids for the His$_6$-tagged recombinant proteins were constructed as follows. Template genomic DNA of M. marisnigri JR-1 ( $=$ ATCC 35101$^T$) was obtained from the American Type Culture Collection (ATCC) and genomic DNAs of M. concilii GP6 ( $=$ NBRC 103675$^T$), M. thermophila ( $=$ NBRC 101360$^T$) and A. profundus

( $=$ NBRC 100127$^T$) were obtained from the NITE Biological Resource Center (NBRC). DNA fragments of sequences encoding PRK homologues were amplified from genomic DNA of M. marisnigri (Mmprk, Memar1934), M. concilii (Mcprk, Mcon1902), M. thermophila (Mtprk, Mthe1447), and A. profundus (Apprk, Arcpr1547) by PCR with the primer sets Mmprk-F and R, Mcprk-F and R, Mtprk-F and R, and Apprk-F and R (Supplementary Table 3), respectively. These DNA fragments with 5'-XhoI and 3'-BamHI sites were inserted into the pET15b vector (Novagen) using an In-Fusion HD Cloning Kit (Takara-Clontech, Japan). The PRK homologues were overexpressed in E. coli C43 (DE3) cells. Induction of gene expression and purification of recombinant proteins were performed using the same procedure as that used for MhPRK.

Expression plasmids for His$_6$-tagged M. hungatei HPS and PHI recombinant proteins were constructed as follows. DNA fragments containing genes encoding HPSs (Mhun0647 encoding MhHPS-MenG and Mhun1628 encoding MhFae-HPS) and PHIs (Mhun0910 encoding MhPHI-a and Mhun3031 encoding MhPHI-b) were amplified from M. hungatei genomic DNA by PCR with the following primer sets: Mhhpsmeng-F and R, Mhfaehps-F and R, Mhphi-a-F and R, and Mhphi-b-F and R (Supplementary Table 3), respectively. These DNA fragments with 5'-XhoI and 3'-BamHI sites were inserted into the pColdI vector (Novagen) using an In-Fusion HD Cloning kit (Takara-Clontech). The recombinant proteins were overexpressed in E. coli C43 (DE3). Cells of E. coli were grown at 37 °C in LB medium containing 50 µg ml$^{-1}$ ampicillin until their optical density at 600 nm reached 0.4–0.6, and then gene expression was induced by adding 1 mM IPTG following a temperature downshift to 15 °C and cultivation at 15 °C for 24 h. After His-tag purification, eluted samples were concentrated using an Amicon Ultra-10 K membrane concentrator (Merck Millipore). Buffer exchange with 50 mM Tris–HCl (pH 8.0) was carried out using a PD-10 column (GE Healthcare).

**PRK activity assay.** PRK activity was assayed at 25 °C with a procedure for a cyanobacterial PRK reported previously[49], with modifications. The reaction mixture (500 µl) contained 100 mM Tris–HCl (pH 8.0), 100 mM KCl, 10 mM MgCl$_2$, 0.2 mM NADH, 2 mM ATP, 2.5 mM phospho(enol)pyruvate, 2 mM Ru5P, 2.5 units L-lactate dehydrogenase (rabbit muscle, Oriental Yeast, Japan), 1 unit pyruvate kinase (rabbit muscle, Oriental Yeast) and purified enzyme. The reaction was initiated by adding the purified enzyme, and the absorbance at 340 nm was monitored with a U3300 spectrophotometer (Hitachi, Japan). Activity was calculated from the molecular extinction coefficient of NADH (6.22 × 10$^3$ M$^{-1}$ cm$^{-1}$ at 340 nm). $V_{max}$ and $K_m$ values were determined from a Lineweaver–Burk plot.

In the assay for substrate specificity of M. hungatei PRK, Ru5P was replaced with the following substrates for P-loop kinases: 6 mM pantothenate, 2 mM thymidine, 2 mM uridine, 2 mM cytidine, 2 mM ribosylnicotinamide, 2 mM fructose 6-phosphate, 2 mM AMP or 2 mM ribose 5-phosphate. Substrate concentrations were selected based on the results of previous studies[14–19].

In the assay for phosphate donor specificity of M. hungatei PRK, ATP was replaced with the following substrates: 2 mM CTP, 2 mM UTP or 2 mM GTP (refs 20,21).

**RuBisCO enzyme assay.** The carboxylase activity of RuBisCO was measured at 25 °C using a procedure modified from Pearce and Andrews[50]. Before the assay, M. hungatei RuBisCO was activated at 37 °C for 30 min in 200 mM HEPES-KOH (pH 8.0) containing 20 mM MgCl$_2$ and 20 mM NaHCO$_3$. The assay mixture (500 µl) contained 200 mM HEPES-KOH (pH 8.0), 20 mM MgCl$_2$, 0.2 mM NADH, 5 mM ATP, 20 mM creatine phosphate, 50 mM NaHCO$_3$, 2 mM RuBP, 11.25 units 3-phosphoglycerate kinase (yeast, Sigma-Aldrich, Japan), 10 units glyceraldehyde-3-phosphate dehydrogenase (rabbit muscle, Sigma-Aldrich), 12.5 units creatine phosphate kinase (rabbit muscle, Oriental Yeast) and 0.05 mg carbonic anhydrase (bovine, Sigma-Aldrich). The assay mixture also included 0.05 units protocatechuate 3,4-dioxygenase (Pseudomonas sp.; Toyobo, Japan) and 4 mM protocatechuate to remove oxygen. All buffers were pre-sparged with N$_2$ and assays were carried out in septum-capped cuvettes with the headspace flushed with N$_2$ to generate anaerobic conditions. The reaction was initiated by adding RuBP, and absorbance at 340 nm was monitored with a U3300 spectrophotometer. Activity was calculated from the molecular extinction coefficient of NADH.

**HPS and PHI activity assays.** HPS and PHI activities were measured at 35 °C using the method of Arfman[51] with modifications. These assays monitored the forward reactions, that is, synthesis toward F6P from Ru5P and formaldehyde. The reaction mixture (500 µl) included 50 mM Tris–HCl (pH 8.0), 5 mM MgCl$_2$, 0.5 mM NADP$^+$, 5 mM Ru5P, 0.5 units glucose-6-phosphate dehydrogenase (yeast, Oriental Yeast), 0.5 units phosphoglucose isomerase (yeast, MP Biomedicals, Germany), 5 mM formaldehyde and purified recombinant enzyme(s).

The reaction was initiated by adding formaldehyde, and absorbance at 340 nm was monitored with a U3300 spectrophotometer. Activity was calculated from the molecular extinction coefficient of NADH. Both HPS and PHI were required in the assay. Therefore, the overall coupling reaction was limited by the amounts of the two enzymes under all reaction conditions.

**Mass spectrometric analysis.** The PRK reaction mixture (1 ml) comprised 50 mM phosphate buffer (pH 8.0), 10 mM $MgCl_2$, 2 mM ATP, 2 mM Ru5P and purified enzyme. The reaction was performed at 37 °C for 60 min, and was terminated by removing enzymes with an Amicon Ultra-0.5 (3 K) centrifugal filter (Merck Millipore). The sample was mixed with methanol to a final concentration of 50% and injected into a Q-TOF Ultima API (Waters, MA, USA) equipped with a nanoflow probe tip (Waters). Spectra were acquired in the negative ion mode in the 50–550 Da mass range. For acquisition of tandem mass spectra, a collision energy of 15 kV was used to induce fragmentation.

**Crystallization and data collection.** MhPRK was expressed in BL21 (DE3) pLysS cells (Novagen), and was purified using Ni-NTA affinity chromatography and gel filtration. Selenomethionine-labelled MhPRK was expressed in B834 (DE3) cells (Novagen) grown in minimal media supplemented with amino acids and selenomethionine, and was purified using a similar protocol as for the native MhPRK. Purified MhPRK and SeMet MhPRK samples were each concentrated to 10 mg ml$^{-1}$. The MhPRK crystals used for X-ray diffraction were grown in 4-μl solutions that contained 2 μl protein solution and 2 μl reservoir solution (20% (w/v) polyethylene glycol 4,000, 0.2 M lithium sulphate, 0.1 M HEPES sodium, pH 7.8), equilibrated against 0.5 ml reservoir solution using the hanging-drop vapour-diffusion method at 20 °C. SeMet-MhPRK crystals used for X-ray diffraction were grown in 2-μl solutions containing 1 μl protein solution and 1 μl reservoir solution (1.5 M ammonium sulphate, 12% (v/v) glycerol, 0.1 M Tris–HCl, pH 8.5), equilibrated against 0.5 ml reservoir solution using the hanging-drop vapour-diffusion method at 20 °C. For both proteins, the crystal that diffracted to the highest resolution was obtained by stirring the crystallization solution[52]. Crystals were each mounted into a loop, and then flash frozen in a stream of nitrogen at 100 K. Diffraction data were collected at 100 K synchrotron radiation at the SPring8 BL44XU (Japan). The diffraction datasets were processed using HKL2000 (ref. 53). Data collection statistics are summarized in Supplementary Table 4.

**Structure determination and refinement.** The MhPRK structure was solved using single-wavelength anomalous diffraction of a SeMet-MhPRK crystal. The structure was refined to 2.5-Å resolution and had $R$ and $R_{free}$ factors of 22.2 and 28.0%, respectively (Supplementary Table 4). The crystallographic asymmetric unit contains two MhPRK protomers (chains A and B). The disordered regions for chain A are the His-tag and residues 1–5, 156–163 and 320–323, while those for chain B are the His-tag and residues 1–3, 156–163 and 320–323. The final model contains 614 amino acid residues and 149 water molecules.

The structure was solved using experimental phases calculated from a single anomalous dispersion experiment. Eighteen possible selenium atoms in an asymmetric unit were found by SHELXD (ref. 54) using the anomalous signals in the SeMet-MhPRK datasets. Initial phases for SeMet-MhPRK were calculated and refined using SHELXE[54] and the graphical interface HKL2MAP (ref. 55) (figure of merit 0.24). The model was refined using CNS (ref. 56) and PHENIX (ref. 57), with manual inspection and modification carried out in conjunction with the CCP4 program COOT (ref. 58). The model had good geometry, with 94.9% of non-glycine residues in the most favoured region and 4.8% in additional allowed regions of the Ramachandran plot as assessed by MolProbity[59]. Refinement statistics are shown in Supplementary Table 4.

Figures were prepared using Pymol (www.pymol.org), Molscript[60] and Raster3D (ref. 61). The final atomic coordinates and structure-factor amplitudes of MhPRK (PDB ID 5B3F) have been deposited in the Worldwide Protein Data Bank (wwPDB; http://www.wwpdb.org) and the Protein Data Bank of Japan at the Institute for Protein Research, Osaka University, Suita, Osaka, Japan (PDBj; http://www.pdbj.org/). Refinement statistics for both coordinate sets are shown in Supplementary Table 4.

**Enzyme assay in cell crude extract.** *M. hungatei* cells cultivated in JCM 242 medium for 2 weeks were harvested by centrifugation (5,000g, 30 min, 4 °C) and cryopreserved at −20 °C. Cells were lysed in 50 mM Tris–HCl (pH 7.5) containing 1 mM EDTA, 2 mM DTT, and protease inhibitor cocktail (Roche Diagnostics, Japan), and disrupted by sonication. The supernatant after centrifugation (20,000g, 30 min, 4 °C) was used as the crude protein extract. The PRK activity in the crude proteins was measured as described above.

Conversion of RuBP or F6P to PGA in extracts was assayed at 37 °C using a modified Sulpice assay[62]. To preactivate RuBisCO, extracts were incubated at 30 °C for 20 min in 100 mM HEPES-KOH (pH 8.0), 5 mM $MgCl_2$, and 50 mM $NaHCO_3$. The treated extract (145 μl) was added to a pre-reaction mixture containing 80 mM HEPES-KOH (pH 8.0), 15 mM $MgCl_2$, 50 mM $NaHCO_3$, 0.05 mg carbonic anhydrase (Sigma-Aldrich), 0.05 units protocatechuate 3,4-dioxygenase (Toyobo), 4 mM protocatechuate, and 5 mM RuBP or F6P in a final volume of 300 μl. The pre-reaction was stopped after 2 h by adding 300 μl 80% (v/v) ethanol. The reaction was initiated by adding 160 μl pre-reacted mixture to 400 μl determination mixture (100 mM HEPES-KOH (pH 8.0), 1.5 mM $MgCl_2$, 0.2 mM NADH, 5 mM ATP, 22.5 units ml$^{-1}$ PGK (Sigma-Aldrich), 20 units ml$^{-1}$ GAPDH (Sigma-Aldrich), 2 units ml$^{-1}$ TPI (Sigma-Aldrich), 1.5 units ml$^{-1}$ G3PDH (Toyobo) and 20 units ml$^{-1}$ G3POX (Toyobo)). The reaction was immediately and continuously

monitored by measuring the absorbance at 340 nm with a U3300 spectrophotometer. To inactivate RuBisCO, CABP was added to extracts at a final concentration of 0.6 mM and then extracts were incubated at 30 °C for 10 min before the reaction. The enzyme activities and amounts of 3-PGA production were calculated from the molecular extinction coefficient of NADH.

**CE–MS analysis of $^{13}$C-labelled *M. hungatei* metabolites.** *In vivo* $^{13}$C-labelling was performed using $NaH^{13}CO_3$ (Cambridge Isotope Laboratories, MA, USA) as the carbon source. *M. hungatei* cells were cultivated in JCM 242 medium for 2 weeks. Four culture vessels (a total of 120 ml of culture, corresponding to approximately 4 mg of cell dry weight) were gas-exchanged and re-pressurized at 150 kPa with 100% $H_2$ gas, and 2 ml of 7.5% $NaH^{13}CO_3$ was then added. After labelling for 1, 3, 5 and 10 min, *M. hungatei* cells were harvested by centrifugation (10,000g, 5 min, 4 °C), washed with 5 ml of ice-cold 50 mM Tris–HCl (pH 8.0), and centrifuged. Harvested cells were suspended in 1 ml of 99.7% methanol (at −30 °C) and disrupted by vortexing for 20 s. After the addition of 400 μl ice-cold $H_2O$ and 1 ml ice-cold chloroform, the suspension was vortexed for 30 s and then centrifuged at 4 °C and 20,000g for 5 min. The supernatant was filtered through an Amicon Ultra-0.5 (3 K) centrifugal filter (Merck Millipore). Internal standards (2 μl of 4 mM L-methionine sulfone, 2 μl of 4 mM PIPES sesquisodium salt, and 2 μl of 10 mM (1 S)-(+)-10-camphorsulfonic acid) were added to 300 μl of the filter sample, which was then chilled with liquid nitrogen and lyophilized. The capillary electrophoresis–mass spectrometric analysis was performed as described previously[40]. Dried metabolites were dissolved in 20 μl of Milli-Q water before CE/MS analysis. The CE/MS experiments were performed using an Agilent G7100 CE system, an Agilent G6224AA LC/MSD timeof-flight (TOF) system, and an Agilent 1,200 series isocratic HPLC pump equipped with a 1:100 splitter for delivery of the sheath liquid. Agilent ChemStation software for CE and MassHunter software for the Agilent TOFMS were used for system control and data acquisition, respectively.

**Data availability.** Atomic coordinates and structure factors for MhPRK have been deposited in the Worldwide Protein Data Bank (wwPDB; http://www.wwpdb.org) and the Protein Data Bank of Japan at the Institute for Protein Research, Osaka University (PDBj; http://www.pdbj.org/) under PDB ID 5B3F. The authors declare that all other data supporting the findings of this study are available within the article and its Supplementary Information files, or from the corresponding authors on request.

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

## Acknowledgements

We thank A. Danchin for valuable help and suggestions, R. Kurata for performing the mass spectrometric analysis, and T. Iino for technical advice on culture of *M. hungatei*. T.K. thanks the Japan Society for the Promotion of Science for providing a JSPS Research Fellowship for Young Scientists. This work was supported by JST, PRESTO, and CREST. This work was also partly supported by Grants-in-Aid for Scientific Research (26292193 to H.A., 17208031 to A.Y. and 25121719, 25440022, 26102526 and 16H00783 to H.M.) and the Program for the Third-Phase R-GIRO Research from the Ritsumeikan Global Innovation Research Organization, Ritsumeikan University. This work has been performed under the approval of the Photon Factory and SPrimg-8 Program Advisory

Committee (Proposal Nos. 2013G148, 2014G685, 2012A6640, 2012B6640, 2013A6848, 2013B6848, 2014A6947, 2014B6947, 2016A6639 and 2016A2572).

## Author contributions

T.K. designed experiments, performed enzymatic analyses for *M. hungatei* PRK and RuBisCO together with S.M., and performed enzyme assays of PHI and HPS. T.K. and C.E. performed activity assays of other archaeal RuBisCOs. N.K., E.M. and H.M. crystallized *M. hungatei* PRK and determined its structure. M.M. and T.H. performed the metabolomic analysis with [13]C labelling of *M. hungatei*. T.I. supervised the structural analysis. H.K. cultured *M. hungatei*. A.Y. and H.A. designed experiments and supervised the study. T.K., H.M. and H.A. wrote the manuscript.

## Additional information

**Competing financial interests**: The authors declare no competing financial interests.

**How to cite this article**: Kono, T. *et al.* A RuBisCO-mediated carbon metabolic pathway in methanogenic archaea. *Nat. Commun.* **8**, 14007 doi: 10.1038/ncomms14007 (2017).

**Publisher's note**: 

