## [Peer Review File · Nature Communications]

Reviewers' comments:

Reviewer #1 (Remarks to the Author):

This paper provides evidence of a novel CO₂ fixation pathway in Archaea, involving two Calvin cycle enzymes Rubisco and PRK, based on phylogenetic analysis, crystallography, enzyme activities and mass spectrometry. These data are interesting and novel providing the first evidence of a partial Calvin cycle outside of photosynthetic organisms. However, most of this work is based on bioinformatic analysis supported by a small amount of biochemistry and therefore can be seen as posing a hypothesis.

The quality of the data presented is satisfactory and supports the majority of the claims however although the data presented provides evidence to suggest that the pathway proposed could be feasible it does not provide sufficient data at this stage to show that in the organism this pathway is actually utilised or has an important role. ¹³C labelling studies together with the creation of knockout mutants (if feasible) would be a way to provide robust data to support the conclusions in this paper. The V_{max} of PRK in these organisms appeared to about 10 fold below that in photosynthetic bacteria and plants, this is not a problem in and of itself but may be suggesting that this is only a minor pathway or may only occur under certain conditions. This is not a problem but needs further explanation before publication.

I am not sure how robust the sequence analysis data is as there seems to be relatively small numbers of e.g. PRK sequences analysed. Given the extent to which this paper is dependent on bioinformatic analysis of sequence this should be extended.

Is there really no transketolase in the Archaea, what about the Oxidative Pentose Phosphate Pathway? is it no present either, this may be the case?

Comments for editorial revisions:

Abstract

1. The Calvin cycle is now usually referred to as the Calvin-Benson Cycle or C₃ cycle
2. The wording 'two enzymes are considered to be unique to the Calvin cycle...is misleading it should say two OF the enzymes, as SBPase is also unique to the C₃ cycle.
3. The sentence beginning 'Here we demonstrate the existence of...' is also open to misinterpretation as the sentence structure could lead the reader to think that Archaea have a C₃ cycle.

Referencing to higher plant C₃ cycle publications is poor e.g. in the Introduction there is no ref cited when the C₃ cycle is discussed, no references to previously published PRK sequence comparisons in higher plants or cyanobacteria. For Rubisco forms only Tabita is cited but Badger and Bek also published on this in 2008 and this paper is more highly cited than Tabita et al 2008. These are just examples and are not a comprehensive review of literature cited.

Reviewer #2 (Remarks to the Author):

The manuscript by Kono et al. describes a novel RuBisCO-dependent CO₂-fixation cycle found mainly in methanogenic Archaea. RuBisCO has been reported previously in Archaea, i.e. *Thermococcus kodakaraensis*, and a function in nucleotide assimilation and degradation has been shown. In this manuscript the second important key enzyme of the photosynthetic Calvin cycle phosphoribulokinase (RPK) was found and characterized in addition to RuBisCO in some mesophilic methanogens and *Archaeoglobus profundus*. Further on the crystal structure of the RPK from *M. hungatei* was solved and the enzyme is compared to the available bacterial photosynthetic structure of *Rhodobacter sphaeroides*. The available knowledge of methanogenic/archaeal central

metabolism was used to propose a new "reductive hexulose phosphate cycle". Finally the co-evolution of PRK and RuBisCO is analyzed and an evolutionary and functional linkage between photosynthetic carbon fixation systems and methanogenesis is proposed.

This study represents an interesting and novel contribution to the field that extends the available knowledge of CO₂ fixation pathways in Archaea and makes an exciting link to the photosynthetic calvin cycle. In general the methodology used is state of the art and the different approaches (including statistics) are suitable and produce good quality of data. However, as discussed below there are some uncertainties regarding the phylogenetic tree construction (see below). The overall presentation of data is attractive. As discussed in detail below regarding the discussion CO₂ fixation cycle or pathway, phylogeny and structure comparison, the manuscript and also the conclusions need some more thoughts and improvement.

According to my knowledge of the field all important previous work has been cited.

The context and of the work is nicely and clearly presented in the abstract and introduction. The Results and Discussion part requires some improvements (see below).

1) The discussion on distribution and phylogeny of PRK (and RuBisCO) in archaea in its present form is improvable: To which archaeal phyla and classes do the PRK/ RuBisCO arboring archaea belong? Does extended data table 2 shows all archaeal PRK representatives? If so, the PRK has a rather narrow distribution in few euryarchaea, and almost exclusively mesophilic methanogens not so deeply branching in the phylogenetic tree of life. No PRK in any crenarchaea, korarchaea or deeply branching Methanopyrus spp.? With respect to this relatively narrow PRK distribution, the archaeal sequences seem far overrepresented in the phylogenetic analyses. Furthermore, do the phylogenies of archaeal PRK as shown in the trees (ex. Data figures 1 and 6) really resemble that of RuBisCO? The PRK and RuBisCO sequence clustering seems to be rather divergent, e.g. the archaeal sequences of *M. tarda*, *M. marisnigri* and *M. boonei* cluster together with hyperthermophiles in the Rubisco tree and with other methanogens in the PRK tree. Does this really indicate coevolution? And also the clustering in archaeal, bacterial and plant types, especially in the RuBisCO tree, appear not that clear as claimed by the authors. The tree shape might also be strongly influenced by the choice of sequences particularly by the overrepresentation of archaeal sequences. Thus, the conclusion that the newly described pathway represents the "evolutionary origin of the photosynthetic Calvin cycle" appears at least over-interpreted.

2) CO₂ fixation pathway or cycle? In my opinion the RHP pathway as proposed by the authors requires some kind of anaplerotic reaction since the CO₂ fixed is withdrawn as HCOH to close the cycle. Hence, if intermediates are taken from the cycle, it would run out of fuel and can thus not act as a metabolic hub as proposed by the authors. In this respect, the connection to the reductive acetyl-CoA pathway and the fate of its product acetyl-CoA should be more carefully discussed. What about e.g. Pyr:Fd OR and PEPS activity in crude extracts. Does the distribution of PRK and Rubisco in archaea coincides with the presence of Fae-HPS? This and hence the interconnection of both, the reductive acetyl-CoA pathway and the newly proposed reductive hexulose phosphate pathway, might be an interesting point to discuss. What is the function of the MenG-HPS fusion? What about the PHIa and b? Is it simply a splitted homolog of other PHI genes? Please comment on this.

3) What about the energetics of the cycle? It is - especially when dealing with methanogens thriving under extreme energy limitation - important to elaborate on this question.

4) The only conclusion the authors' state from the crystal structure of PRK is that it is really a PRK. Is that all to get out of that? What about the structural differences compared to the *R. sphaeroides* PRK, what is the impact of the divergent C-terminus and the different oligomerization?

5) Species names in italics

Reviewer #3 (Remarks to the Author):

The manuscript, A RuBisCO-participating, novel CO₂ fixation cycle in methanogenic Archaea, reports on the discovery of a new carbon dioxide fixing pathway in the methanogen

Methanospirillum hungatei. Carbon is fixed in the RuBisCO reaction, while formaldehyde is released at the arabino-3-hexulose-6-phosphate synthase (HPS) reaction using fructose 6-phosphate (F6P) as a substrate. The ribulose 5-phosphate (Ru5P) generated by HPS is converted back to ribulose 1,5-bisphosphate (RuBP) by a newly identified archaeal phosphoribulokinase. As 3-phosphoglycerate, the product of the RuBisCO reaction, can be converted to F6P via gluconeogenesis, the authors propose that these enzymes constitute a novel carbon dioxide fixing cycle.

The manuscript can be divided into three parts, in terms of the experimental results. The first is the identification and biochemical characterization of the first phosphoribulokinase in archaea. The second is the structural examination of the phosphoribulokinase. The third is the search/identification of enzymes that can supply Ru5P, the substrate for phosphoribulokinase. The first two parts are important findings that should significantly contribute to our understanding of Rubisco-related metabolism and archaeal metabolism in general. The third part reveals what enzymes are responsible for providing Ru5P. This provides the final link that connects F6P to 3-PGA via Ru5P and RuBP. In addition, the presence of a formaldehyde activating enzyme fused to HPS suggests that the cycle is linked to methanogenesis and the reductive acetyl-CoA pathway. The three parts of the manuscript reveal a new metabolic route in a number of methanogens and is of high interest.

One point that should be considered is the designation of the pathway as a carbon dioxide fixing pathway. First of all, the material balance of the cycle should clearly be stated. If correct, is it that one molecule of carbon dioxide is reduced to one molecule of formaldehyde (using 2 ATP molecules and 1 NADPH)?

Next, the generated product should be able to act as a precursor for all cell material. Is this the Fae-HPS connection to acetyl-CoA (and pyruvate)? In this sense, shouldn't the fixation pathway also include the steps from formaldehyde release to acetyl-CoA generation? Please comment. The cycle alone is somewhat reminiscent of the 3-hydroxypropionate pathway prior to the discovery of how glyoxylate was converted to cell material. Other links to central carbon metabolism should be distinguished from the main pathway, as they would consume the intermediates of the cycle and these would have to be replenished by anaplerotic reactions. The most important of these would seem to be the link from acetyl-CoA to pyruvate, PEP, 2-phosphoglycerate and finally 3-PGA. Are these enzymes present?

The third point is whether the metabolism can support autotrophic growth. This is difficult to prove, as the route from formaldehyde to acetyl-CoA/pyruvate is shared with the reductive acetyl-CoA pathway. The pathway may well serve as a central metabolic hub, and the route from F6P to 3-PGA can also be considered a clever strategy to minimize carbon loss via the RuMP pathway. Please comment. Do all methanogens harboring this cycle possess the HPS fused with Fae?

Reference 6 is probably Reference 8

Thermococcus kodakaraensis should now be referred to as *Thermococcus kodakarensis*

Although a number of substrates have been tested, does the PRK protein exhibit activity towards ribulose? Is ATP the only phosphate donor recognized?

Pg 4, 3rd line from bottom: Delete one of the two *M. hungatei*.

Pg 5, Line 11, Concerning transketolase, many archaea are known not to harbor a transaldolase. Does this organism harbor a transaldolase? If this archaeon does not possess a transketolase, how does it synthesize aromatic amino acids, or folate (via chorismate)?

Pg 5, Lines 19-28, Although stated in the methods section, this section (and others) should be written in more detail. The HPS-MenG and Fae-HPS activities; are these the activity levels of purified, recombinant proteins? What activities are the authors measuring, and in which direction?

Is the FBPase in this archaeon a distinct enzyme from the aldolase, or is it an archaeal-type fused FBPase/FBP aldolase?

Reviewer #4 (Remarks to the Author):

The authors report the identification of a new CO₂ fixation cycle, which is present in methanogenic archaea. They found a homolog of photosynthetic phosphoribulokinase (PRK) in methanogenic archaea, and confirmed its catalytic activity. A crystal structure at 2.5 Å resolution of the PRK from *M. hungatei* (MhPRK) showed that it is very similar to that of photosynthetic bacterial PRK (RsPRK), consistent with ~30% sequence identity between them.

I will focus on the structural studies in this report, and have only a few minor comments -

1. The dimer and quaternary structure of MhPRK is different from RsPRK (p. 4, Extended Data Fig. 4). Does this have any functional implications?
2. Strands b7 and b8 in the structure of RsPRK should be named b8 and b9, as they are topologically equivalent to these two strands in MhPRK. What is the rms distance in Ca positions when the two structures are overlaid?
3. The Molprobit analysis suggests 1% of the residues are in the disallowed region of the Ramachandran plot. This is fairly high and should be corrected/justified.

Responses to Reviewer #1

Comments from Reviewer #1

This paper provides evidence of a novel CO₂ fixation pathway in Archaea, involving two Calvin cycle enzymes Rubisco and PRK, based on phylogenetic analysis, crystallography, enzyme activities and mass spectrometry. These data are interesting and novel providing the first evidence of a partial Calvin cycle outside of photosynthetic organisms. However, most of this work is based on bioinformatic analysis supported by a small amount of biochemistry and therefore can be seen as posing a hypothesis.

The quality of the data presented is satisfactory and supports the majority of the claims however although the data presented provides evidence to suggest that the pathway proposed could be feasible it does not provide sufficient data at this stage to show that in the organism this pathway is actually utilised or has an important role. ¹³C labelling studies together with the creation of knockout mutants (if feasible) would be a way to provide robust data to support the conclusions in this paper. The V_{max} of PRK in these organisms appeared to about 10 fold below that in photosynthetic bacteria and plants, this is not a problem in and of itself but may be suggesting that this is a only a minor pathway or may only occur under certain conditions. This is not a problem but needs further explanation before publication.

Our response:

Thank you for important suggestions to strengthen our ideas in this manuscript.

Reviewer #1 proposed that ¹³C-labelling experiment together with creation of knockout mutants of *M. hungatei*. Unfortunately, a transformation method has not been established in this Archaeon as well as many other methanogenic Archaea. Therefore we performed metabolome analysis coupled with ¹³C-labelling to follow incorporated carbons and analyze the novel CO₂-fixing pathway in wild-type *M. hungatei* cells.

Incorporation of ¹³C atoms was detected in metabolites, with particularly high ¹³C-labelling rates observed for 3-PGA, FBP, F6P, G6P, 2-PGA, and PEP (Fig. 4 and Supplementary Fig. 9). Considering that the RuBisCO was active in extracts and labelling rate and pool size of 3-PGA were high, 3-PGA was expected to be the first major compound with incorporated ¹³CO₂ in *M. hungatei*. These results support the idea that incorporation of ¹³C atoms into 3-PGA mainly stemmed from the carboxylase

reaction of RuBisCO. FBP and F6P showed high ^{13}C labelling rates and Ru5P showed a significant, albeit lower, one, strongly suggesting that the RHP pathway was active in *M. hungatei*. However, the ^{13}C -atom labelling rate and pool size of 3-PGA were much lower than that observed in photosynthetic organisms. This result suggests that investment of energy in the RHP pathway is much lower than what plant and cyanobacteria invest in the Calvin-Benson cycle. Methanogenic Archaea carry out methanogenesis with a low energy output compared with photosynthesis, a conclusion consistent with the observation that *M. hungatei* RuBisCO and PRK activities were much lower than those of photosynthetic organisms.

Data of metabolomic analysis were added as new Figure 4, Supplementary Figure 9, and Supplementary Figure 10.

Above results and discussions were added in result and revised manuscript.

Comments from Reviewer #1

I am not sure how robust the sequence analysis data is as there seems to be relatively small numbers of e.g. PRK sequences analysed. Given the extent to which this paper is dependent on bioinformatic analysis of sequence this should be extended.

Our response:

We reanalyzed PRK sequences with extended sequence number. In the result, PRKs were more clearly divided into three clades, Plant-type, photosynthetic bacterial type and Archaeal type. The new phylogenetic tree was added as a new Supplementary Figure 1 in the revised manuscript and above result added in “Enzymatic analysis of archaeal PRK homologues” in the results section (Page 3).

Comments from Reviewer #1

Is there really no transketolase in the Archaea, what about the Oxidative Pentose Phosphate Pathway? is it no present either, this may be the case?

Our response:

In general, Archaea do not possess transketolase and also transaldolase. Therefore, it has been expected that the Oxidative Pentose Phosphate Pathway does not work in

Archaea. We added above information in “Putative novel CO₂-fixing pathway involving PRK and RuBisCO in *M. hungatei*” in the results section (page 6) of the revised text.

Comments for editorial revisions:

Abstract

1. The Calvin cycle is now usually referred to as the Calvin-Benson Cycle or C3 cycle

Our response:

As Reviewer #1 suggested, we changed from the “Calvin cycle” to the “Calvin-Benson cycle” in the revised manuscript.

2. The wording 'two enzymes are considered to be unique to the Calvin cycle...is misleading it should say two OF the enzymes, as SBPase is also unique to the C3 cycle.

Our response:

The sentence made a misleading in the original manuscript, as suggested by Reviewer #1. We changed from “two enzymes” to “two of enzymes” in the revised manuscript.

3. The sentence beginning 'Here we demonstrate the existence of... is also open to misinterpretation as the sentence structure could lead the reader to think that Archaea have a C3 cycle.

Our response

In order to avoid leading misinterpretation, “similar to the photosynthetic Calvin cycle” has been removed from the original sentence in the revised manuscript.

Referencing to higher plant C3 cycle publications is poor e.g. in the Introduction there is no ref cited when the C3 cycle is discussed, no references to previously published PRK sequence comparisons in higher plants or cyanobacteria. For Rubisco forms only Tabita is cited but Badger and Bek also published on this in 2008 and this paper is more highly cited than Tabita et al 2008. These are just examples and are not a comprehensive review of literature cited.

Our response:

We added references respect to the Calvin-Benson cycle, PRK comparison, RuBisCO forms, and others in the revised manuscript.

Reviewer #2:

The manuscript by Kono et al. describes a novel RuBisCO-dependent CO₂-fixation cycle found mainly in methanogenic Archaea. RuBisCO has been reported previously in Archaea, i.e. *Thermococcus kodakaraensis*, and a function in nucleotide assimilation and degradation has been shown. In this manuscript the second important key enzyme of the photosynthetic calvin cycle phosphoribulokinase (RPK) was found and characterized in addition to RuBisCO in some mesophilic methanogens and *Archaeoglobus profundus*. Further on the crystal structure of the RPK from *M. hungatei* was solved and the enzyme is compared to the available bacterial photosynthetic structure of *Rhodobacter sphaeroides*. The available knowledge of methanogenic/archaeal central metabolism was used to propose a new "reductive hexulose phosphate cycle". Finally the co-evolution of RPK and RuBisCO is analyzed and an evolutionary and functional linkage between photosynthetic carbon fixation systems and methanogenesis is proposed.

This study represents an interesting and novel contribution to the field that extends the available knowledge of CO₂ fixation pathways in Archaea and makes an exciting link to the photosynthetic calvin cycle. In general the methodology used is state of the art and the different approaches (including statistics) are suitable and produce good quality of data. However, as discussed below there are some uncertainties regarding the phylogenetic tree construction (see below). The overall presentation of data is attractive. As discussed in detail below regarding the discussion CO₂ fixation cycle or pathway, phylogeny and structure comparison, the manuscript and also the conclusions need some more thoughts and improvement.

According to my knowledge of the field all important previous work has been cited. The context and of the work is nicely and clearly presented in the abstract and introduction. The Results and Discussion part requires some improvements (see below).

1) The discussion on distribution and phylogeny of PRK (and RuBisCO) in archaea in its present form is improvable: To which archaeal phyla and classes do the PRK/RuBisCO harboring archaea belong? Does extended data table 2 shows all archaeal PRK representatives? If so, the PRK has a rather narrow distribution in few euryarchaea, and almost exclusively mesophilic methanogens not so deeply branching in the

phylogenetic tree of life. No PRK in any crenarchaea, korarchaea or deeply branching Methanopyrus spp.? With respect to this relatively narrow PRK distribution, the archaeal sequences seem far overrepresented in the phylogenetic analyses. Furthermore, do the phylogenies of archaeal PRK as shown in the trees (ex. Data figures 1 and 6) really resemble that of RuBisCO? The PRK and RuBisCO sequence clustering seems to be rather divergent, e.g. the archaeal sequences of *M. tarda*, *M. marisnigri* and *M. boonei* cluster together with hyperthermophiles in the Rubisco tree and with other methanogens in the PRK tree. Does this really indicate coevolution? And also the clustering in archaeal, bacterial and plant types, especially in the RuBisCO tree, appear not that clear as claimed by the authors. The tree shape might also be strongly influenced by the choice of sequences particularly by the overrepresentation of archaeal sequences. Thus, the conclusion that the newly described pathway represents the "evolutionary origin of the photosynthetic Calvin cycle" appears at least over-interpreted.

Our response:

Thank you so much for your important suggestions. PRKs were found in only fifteen Archaea species so far. Therefore, the extended data table 2 (supplementary table 1 in the revised manuscript) contains all Archaea species possessing PRK. PRKs were found in only Euryarchaeota, not in Crenarchaeota, Thaumarchaeota, Korarchaeota. Therefore, Archaea species possessing PRK genes are limited in Euryarchaeota so far. In order to discuss as careful as possible using limited Archaeal PRK sequences, we reanalyzed a phylogenetic tree of PRK by increasing PRK sequences from photosynthetic organisms. This phylogenetic tree was shown as a new Supplementary Figure 1 in the revised manuscript. In new phylogenetic tree, PRKs were more clearly divided into three clades, Plant-type, photosynthetic bacterial type and Archaeal type. In addition, a phylogenetic tree of RuBisCO was also reanalyzed by addition of more sequences. This phylogenetic tree of RuBisCO was shown as a new Supplementary Figure 8. The phylogenetic tree of PRK is different from that of RuBisCO, as suggested by reviewer #2. Therefore, we modified discussion on phylogeny of PRK as below;

Page 5, lines 34–36,

On the other hand, archaeal PRK forms a clade that is clearly separated from bacterial

and plant-type clades (Supplementary Fig. 1), which suggests that the molecular evolution of RuBisCO and PRK has been different.

2) CO₂ fixation pathway or cycle? In my opinion the RHP pathway as proposed by the authors requires some kind of anaplerotic reaction since the CO₂ fixed is withdrawn as HCOH to close the cycle. Hence, if intermediates are taken from the cycle, it would run out of fuel and can thus not act as a metabolic hub as proposed by the authors. In this respect, the connection to the reductive acetyl-CoA pathway and the fate of its product acetyl-CoA should be more carefully discussed. What about e.g. Pyr:Fd OR and PEPS activity in crude extracts. Does the distribution of PRK and Rubisco in archaea coincides with the presence of Fae-HPS? This and hence the interconnection of both, the reductive acetyl-CoA pathway and the newly proposed reductive hexulose phosphate pathway, might be an interesting point to discuss. What is the function of the MenG-HPS fusion? What about the PHIa and b? Is it simply a splitted homolog of other PHI genes? Please comment on this.

Our responses:

As suggested by reviewer #2, the carbon fixed by RuBisCO is reduced and released as formaldehyde in our proposed metabolism. Therefore, it is better that we name this metabolism, “the RHP pathway”. We have modified from RHP cycle to RHP pathway in the revised manuscript.

New additional data of ¹³C-labelling metabolomic analysis revealed that the RHP pathway is active in vivo and that carbons fixed by RuBisCO were also effluxed to gluconeogenesis and glycolysis. This fact also supports that we had better to use “pathway”. This result indicated that the RHP pathway dose not act as a metabolic hub too, as pointed out by reviewer #2. Therefore, we removed the word “metabolic hub” in the revised manuscript. Results of ¹³C-labelling metabolomic analysis were added in “Putative novel CO₂-fixing pathway involving PRK and RuBisCO in *M. hungatei*” in the results section (page 7) and as Figure 4, Supplementary Figures 9 and 10 in the revised manuscript.

We think that the connection of the RHP pathway to the reductive acetyl-CoA pathway is important. In our metabolomic analysis data, ¹³C atoms were incorporated into acetyl-CoA at a low rate, and therefore the reductive acetyl-CoA pathway might be

active in vivo. However, our study could not follow carbons from the RHP pathway to acetyl-CoA, because we do not have a metabolomic analysis system to detect intermediates specific in the reductive acetyl-CoA pathway. We would like to access to this issue as a future works.

Incorporation of ^{13}C atoms was not detected in pyruvate at all, which suggested that Pyr:Fd OR and PEPS work with very low activity or do not work. The fate of acetyl-CoA, product of the reductive acetyl-CoA pathway, should not turn toward to the RHP cycle via PGA.

Archaea having both RuBisCO and PRK, except *Aciduliprofundum boonei*, possess genes for Fae and HPS or their fused protein. Demethylmenaquinone methyltransferase (MenG) catalyses the last step in menaquinone biosynthesis. We have no idea for a functional linkage between HPS and MenG-like domain at present time. Therefore, the function of MenG-like domain is unknown in the MenG-HSP fusion protein. *M.*

hungatei possesses two genes for PHI. We named two PHI as PHI-a and PHI-b.

Above information and discussion were added in “Putative novel CO_2 -fixing pathway involving PRK and RuBisCO in *M. hungatei*” in the results section (pages 7–8) in the revised manuscript.

3) What about the energetics of the cycle? It is - especially when dealing with methanogens thriving under extreme energy limitation - important to elaborate on this question.

Our responses: As suggested by reviewer #2, methanogen utilizes the methanogenesis with a low energetic output. If we consider a cycle of the RHP pathway, one molecule of CO_2 reduced to one molecule of formaldehyde using three molecules of ATP and two molecules of NAD(P)H. Therefore, the cycle may require high cost. The energy yield of methanogenesis under standard conditions is 131 kJ/mol. Under ecological conditions of 1–10 Pa H_2 , the energy yield is lower 15–35 kJ/mol. This is insufficient for synthesis of 1 molecule of ATP per 1 molecule of methane produced. The methanogenesis coupled with ATP synthesis depends on substrate H_2 concentrations. Therefore, the cycle may work more in high H_2 conditions. Actually, *M. hungatei* cells used for our experiments were grown in high H_2 condition (150 KPa with H_2). On the other hand, supply of formaldehyde from the RHP pathway enables to skip three steps of

methanogenesis including the first CO₂ reduction step, catalyzed by formyl-methanofuran dehydrogenase, which is the only endergonic reaction. This shortcut may increase an efficiency of ATP production. However, the energetic of the RHP pathway in the low-energy lifestyle of methanogen remains to be further addressed.

Above discussion with respect to the energetics of the RHP pathway was added in the discussion section (page 8) in the revised manuscript.

4) The only conclusion the authors' state from the crystal structure of PRK is that it is really a PRK. Is that all to get out of that? What about the structural differences compared to the *R. sphaeroides* PRK, what is the impact of the divergent C-terminus and the different oligomerization?

Our response:

We could not discuss about indications from the crystal structure of *M. hungatei* PRK, as reviewer #2 suggested.

In dimeric *M. hungatei* PRK, the central strands $\beta 7$, which consolidates the formation of the dimer with the larger dimer interface (1694.6 Å²), and $\beta 8$ form the dimer interface. In *R. sphaeroides* PRK, strands $\beta 5$, 6 and 9 and α -helix 6 participate in the formation of the dimer interface, and α -helix 7 is involved in octamer formation. These structures involved in dimer and octamer formation are placed in the C-terminal domain. The N-terminal domain (1–198) of *M. hungatei* PRK resembles that of *R. sphaeroides* PRK, whereas the C-terminal domain (199–319 for *M. hungatei* PRK) is relatively different (Supplementary Fig. 2). Therefore, a low similarity in the C-terminal domain between *M. hungatei* PRK and *R. sphaeroides* PRK is consistent with differences in the manner of dimerization and quaternary structure of the two, which may lead to different enzymatic properties such as allosteric regulation of *R. sphaeroides* PRK, and not *M. hungatei* PRK by NADH and AMP.

Above discussion was added in “Crystal structure of *M. hungatei* PRK” in the results section (page 5) in the revised manuscript.

5) Species names in italics

Our response:

Species names were collected in italics in the revised manuscript.

Reviewer #3 (Remarks to the Author):

The manuscript, A RuBisCO-participating, novel CO₂ fixation cycle in methanogenic Archaea, reports on the discovery of a new carbon dioxide fixing pathway in the methanogen *Methanospirillum hungatei*. Carbon is fixed in the RuBisCO reaction, while formaldehyde is released at the arabino-3-hexulose-6-phosphate synthase (HPS) reaction using fructose 6-phosphate (F6P) as a substrate. The ribulose 5-phosphate (Ru5P) generated by HPS is converted back to ribulose 1,5-bisphosphate (RuBP) by a newly identified archaeal phosphoribulokinase. As 3-phosphoglycerate, the product of the RuBisCO reaction, can be converted to F6P via gluconeogenesis, the authors propose that these enzymes constitute a novel carbon dioxide fixing cycle.

The manuscript can be divided into three parts, in terms of the experimental results. The first is the identification and biochemical characterization of the first phosphoribulokinase in archaea. The second is the structural examination of the phosphoribulokinase. The third is the search/identification of enzymes that can supply Ru5P, the substrate for phosphoribulokinase. The first two parts are important findings that should significantly contribute to our understanding of Rubisco-related metabolism and archaeal metabolism in general. The third part reveals what enzymes are responsible for providing Ru5P. This provides the final link that connects F6P to 3-PGA via Ru5P and RuBP. In addition, the presence of a formaldehyde activating enzyme fused to HPS suggests that the cycle is linked to methanogenesis and the reductive acetyl-CoA pathway. The three parts of the manuscript reveal a new metabolic route in a number of methanogens and is of high interest.

One point that should be considered is the designation of the pathway as a carbon dioxide fixing pathway. First of all, the material balance of the cycle should clearly be stated. If correct, is it that one molecule of carbon dioxide is reduced to one molecule of formaldehyde (using 2 ATP molecules and 1 NADPH)?

Our response:

One molecule of carbon dioxide is reduced to one molecule of formaldehyde, which requires two molecules of both ATP and NAD(P)H. In the case including PRK reaction for cyclization of the pathway, it is required of another one molecule of ATP. This means that one molecule of CO₂ convert to one molecule of formaldehyde using three molecules of ATP and two molecules of NADH. These states were added in the discussion section (page 8) in the revised manuscript.

Next, the generated product should be able to act as a precursor for all cell material. Is this the Fac-HPS connection to acetyl-CoA (and pyruvate)? In this sense, shouldn't the fixation pathway also include the steps from formaldehyde release to acetyl-CoA generation? Please comment. The cycle alone is somewhat reminiscent of the 3-hydroxypropionate pathway prior to the discovery of how glyoxylate was converted to cell material. Other links to central carbon metabolism should be distinguished from the main pathway, as they would consume the intermediates of the cycle and these would have to be replenished by anaplerotic reactions. The most important of these would seem to be the link from acetyl-CoA to pyruvate, PEP, 2-phosphoglycerate and finally 3-PGA. Are these enzymes present?

Our response:

New ¹³C-labelling Metabolomic analysis suggested that ¹³C atoms were incorporated into acetyl-CoA, suggesting that the reductive acetyl-CoA pathway might be active in vivo. However, our study could not follow carbon from formaldehyde release to acetyl-CoA. Because, we did not have a metabolomic analysis system for metabolites, such as methylene tetrahydromethanopterin, that are specific for this pathway. Therefore, it is better that the RHP pathway does not include the reductive acetyl-CoA pathway at the present situation. We discussed about a metabolic link between the RHP and reductive acetyl-CoA pathways, as a possibility.

As Reviewer #3 suggested, the RHP cycle requires anaplerotic reactions because carbons were flowed out from the RHP cycle as formaldehyde and intermediates for other metabolic pathways. Actually, our metabolomics analysis revealed that a part of materials flowed from the RHP pathway to gluconeogenesis and glycolysis. *M. hungatei* possesses all genes for enzymes catalyzing steps from acetyl-CoA to 3PGA (pyruvate

ferredoxin oxidoreductase, *Mhun0450* (γ subunit), *0451* (δ subunit), *0452* (α subunit) and *0453* (β subunit); phosphoenolpyruvate synthase, *Mhun2610*; phosphopyruvate hydratase, *Mhun1018*, *1101* and *2893*; phosphoglycerate mutase, *Mhun0447* and *2324*). In metabolomic analysis data, ^{13}C atoms were incorporated into acetyl-CoA, but not into pyruvate at all, suggesting that the carbons were not replenished from acetyl-CoA to RHP pathway. *M. hungatei* conserves genes for the pentose bisphosphate pathway to generate RuBP for RuBisCO from nucleoside. Therefore, the pentose bisphosphate pathway may serve as the anaplerotic metabolism for the RHP pathway. These information and discussion were added in the discussion section (page 8 and 9) in the revised manuscript.

The third point is whether the metabolism can support autotrophic growth. This is difficult to prove, as the route from formaldehyde to acetyl-CoA/pyruvate is shared with the reductive acetyl-CoA pathway. The pathway may well serve as a central metabolic hub, and the route from F6P to 3-PGA can also be considered a clever strategy to minimize carbon loss via the RuMP pathway. Please comment. Do all methanogens harboring this cycle possess the HPS fused with Fae?

Our response:

We expect that the RHP pathway supports autotrophic growth by CO_2 fixation. Actually, the carbons fixed by RuBisCO were supplied for gluconeogenesis and glycolysis in metabolomic data. In the case of RuBP regeneration in the RHP pathway, formaldehyde released from this pathway could be recovered as a carbon source by the reductive acetyl-CoA pathway. In addition, formaldehyde can be also utilized for the methanogenesis to produce energy. These can minimize carbon loss and recover a part of energy, as Reviewer #3 suggested. Archaea harbouring genes for the RHP pathway, except *Aciduliprofundum boonei*, possess genes for Fae and HPS (*Ferroglobus placidus*, *Methanoculleus marisnigri*, *Methanosaeta concilii* and *Methanosaeta harundinacea*) or their fused protein (other 10 species). These facts indicate that formaldehyde released from the RHP pathway recovered by the reductive acetyl-CoA pathway and the methanogenesis.

Above discussion was added in discussion section (page 9) in the revised manuscript.

Reference 6 is probably Reference 8

Our Response:

Reference was modified in the revised manuscript.

Thermococcus kodakaraensis should now be referred to as *Thermococcus kodakarensis*

Our Response:

We modified from “*Thermococcus kodakaraensis*” to “*Thermococcus kodakarensis*”.

Although a number of substrates have been tested, does the PRK protein exhibit activity towards ribulose? Is ATP the only phosphate donor recognized?

Our Response:

It has not been reported that photosynthetic PRKs utilize substrates other than ribulose-5-phosphate as a phosphate acceptor. *M. hungatei* PRK was specific for ribulose, in our result. We performed new experiment to analyze specificity for phosphate donor substrates of *M. hungatei* PRK. *M. hungatei* PRK utilized broad phosphate donor substrates, such as ATP, GTP, CTP, and UTP. Kinase activities with CTP, UTP, and GTP were respectively 74.35%, 60.48%, and 93.96% of that with ATP. This result was new and added as Supplementary Figure 4. On the other hand, photosynthetic PRK as a phosphate donor is relatively specific to ATP.

Above results and information was added in “Enzymatic analysis of archaeal PRK homologues“ in the results section (page 4) and as Supplementary Figure 4 in the revised manuscript.

Pg 4, 3rd line from bottom: Delete one of the two *M. hungatei*.

Our Response:

We modified the original manuscript, as suggested by reviewer #3.

Pg 5, Line 11, Concerning transketolase, many archaea are known not to harbor a transaldolase. Does this organism harbor a transaldolase? If this archaeon does not possess a transketolase, how does it synthesize aromatic amino acids, or folate (via chorismate)?

Our Response:

Archaea having both RuBisCO and PRK including *M. hungatei*, do not possess genes for transketolase and transaldolase needed for the oxidative pentose phosphate pathway; instead, aromatic amino acids are synthesized via chorismate and shikimate in the 6-deoxy-5-ketofructose-1-phosphate pathway. Above information was added in “Putative novel CO₂-fixing pathway involving PRK and RuBisCO in *M. hungatei*” in the results section (page 6) in the revised manuscript.

Pg 5, Lines 19-28, Although stated in the methods section, this section (and others) should be written in more detail. The HPS-MenG and Fae-HPS activities; are these the activity levels of purified, recombinant proteins? What activities are the authors measuring, and in which direction?

Our Response:

The purified *E. coli* recombinant proteins were used for enzyme assay, and measured the fixing reaction of the formaldehyde. These assays monitored the forward reactions, i.e., synthesis toward F6P from Ru5P and formaldehyde. HPS is a reversible enzyme and therefore the HPS activities were measured more easily detected direction. Above information was added in “HPS and PHI activity assays” in the “Methods” section of revised manuscript (pages 14 and 15).

Is the FBPase in this archaeon a distinct enzyme from the aldolase, or is it an archaeal-type fused FBPase/FBP aldolase?

Our response:

In *M. hungatei*, FBPase and FBP aldolase are separated.

Reviewer #4 (Remarks to the Author):

The authors report the identification of a new CO₂ fixation cycle, which is present in methanogenic archaea. They found a homolog of photosynthetic phosphoribulokinase (PRK) in methanogenic archaea, and confirmed its catalytic activity. A crystal structure at 2.5 Å resolution of the PRK from *M. hungatei* (MhPRK) showed that it is very similar to that of photosynthetic bacterial PRK (RsPRK), consistent with ~30% sequence identity between them.

I will focus on the structural studies in this report, and have only a few minor comments

-

1. The dimer and quaternary structure of MhPRK is different from RsPRK (p. 4, Extended Data Fig. 4). Does this have any functional implications?

Our response:

Yes, it is an important point. MhPRK and RsPRK adopt as dimer and octamer, respectively. We cannot rule out the possibility that the dimeric architecture of MhPRK might reflect the susceptibility to a variety of environmental changes. In previous reports, RsPRK is allosteric regulated by NADH and AMP, however MhPRK is not at all (data not shown). Difference of quaternary structure may be involved in this distinct enzymatic property. Above discussion was added in “Crystal structure of *M. hungatei* PRK” of result section (Page 5) in the revised manuscript.

2. Strands b7 and b8 in the structure of RsPRK should be named b8 and b9, as they are topologically equivalent to these two strands in MhPRK. What is the rms distance in C α positions when the two structures are overlaid?

Our response:

Thank you for your suggestion. We renamed strands b7 and b8 of RsPRK to b8 and b9 in the main text and Figure 1 b and d. We also added rms distance in C α position when the two structures are overlaid in “Crystal structure of *M. hungatei* PRK” of result section (Page 4).

3. The Molprobit analysis suggests 1% of the residues are in the disallowed region of the Ramachandran plot. This is fairly high and should be corrected/justified.

Our response:

Thank you for your suggestion. We corrected the crystal structure, and deposited the new coordinates in protein data bank. We accordingly revised the crystallographic table (Supplementary Table 3) and main text (“Structure determination and refinement” in the Methods section, Page 16).

Reviewers' comments:

Reviewer #1 (Remarks to the Author):

My one comment would be for the authors to consider if the phylogenetic analysis of PRK and Rubisco like proteins should be in the main body of the paper.

Reviewer #2 (Remarks to the Author):

Although the authors tried to improve their manuscript "A RuBisCO-participating, novel CO₂ fixation pathway in methanogenic archaea" by several efforts including ¹³C labelling studies, reanalysing phylogenies etc., the major question i.e. what is the fate of formaldehyde is so far - in my opinion - not satisfactory answered. Without that information the proposed pathway is far from being a complete CO₂ fixation pathway. It just provide some evidence that PRK and RuBisCO are involved in biomass synthesis under autotrophic conditions.

Formaldehyde, the product of the RHP pathway, has to be fixed itself into biomass otherwise the story would not make any sense. If the formaldehyde channeled into methanogenesis as suggested as one possibility by the authors CO₂ would be reduced to formaldehyde at the expense of three ATP and then would be further reduced to methane yielding roughly only 1 ATP, even under high H₂ partial pressure this will never end up in a positive energy balance.

The second, more plausible, explanation would be a concerted action of the proposed RHP pathway and the reductive acetyl-CoA pathway. Then, one CO₂ is fixed in the RHP pathway yielding one molecule of formaldehyde which is then converted via the reductive acetyl-CoA pathway, Pyr:Fd OR, PEP synthetase finally to 3-PGA. Such a pathway would result in a net yield of three fixed CO₂ molecules into one molecule of 3-PGA which would keep the cycle running. Also, the energetics would be more efficient: The fixation of three CO₂ into one 3-PGA would cost roughly 4 ATP which is in the same range as the reductive acetyl-CoA pathway, the energetically most effective CO₂ fixation pathway described so far. However, the results presented by the authors appear to argue against this possibility, and thus the manuscript still lacks an explanation how the proposed metabolism would not run out of fuel and how the energetic balance becomes positive. The suggestion that the pentose bisphosphate pathway might act as anaplerotic sequence appears also not valid since also nucleosides have to be synthesised at last from CO₂ under autotrophic conditions, but an explanation how this is achieved is not given.

Altogether, the authors need in my opinion to provide a reasonable explanation how formaldehyde is converted into biomass.

Reviewer #3 (Remarks to the Author):

The revised version of the manuscript, A RuBisCO-participating, novel CO₂ fixation cycle in methanogenic Archaea, has adequately addressed and discussed the comments suggested by reviewer #3. The reviewer has only several minor/trivial points for consideration.

1. Pg2, line 1:

Two of the enzymes should read Two enzymes

2. Pg3, lines 5-6: to generate RuBP from NMP for RuBisCO has been especially well characterized can be modified to

to generate RuBP from nucleosides or NMP for RuBisCO has been especially well characterized
Same comment for Pg 9, line 4.

This can also be incorporated into the top of Figure 2, but I leave this to the authors' judgment.

3. Pg4, line 9 from bottom: Can the authors insert the identity here?
in spite of their low amino acid identity (xx%).

4. Pg 6, line 10: Archaea do not possess genes for transketolase and transaldolase needed for the oxidative pentose phosphate pathway
I think the authors are referring to the non-oxidative pentose phosphate pathway here.

5. Supplementary Figure 1, Legend lines 1&2: phylogenetic

6. Supplementary Figure 8, Legend
lines 1&2: phylogenetic

line 2: is the multiple sequence alignment intended to be shown or is this just referring to the process of tree construction?

Species abbreviations: are only those not designated in Supplementary Figure 1 written here?

7. Supplementary Table 1: should read CO dehydrogenase

8. Table 1: should read 20.7 ± 1.7

9. Pg 11, Line 5, The authors refer to bootstrap analyses here. Either include values in the trees or modify the text.

Reviewer #4 (Remarks to the Author):

The authors have addressed my concerns regarding the crystal structure.

Reviewers' comments:

Reviewer #1 (Remarks to the Author):

My one comment would be for the authors to consider if the phylogenetic analysis of PRK and Rubisco like proteins should be in the main body of the paper.

Our response:

Thank you for your important suggestion. The phylogenetic trees of PRK (Supplementary Figure 1) and RuBisCO (Supplementary Figure 8) were moved from the supplementary information to the main text as Figure 1 and Figure 3, respectively.

Reviewer #2 (Remarks to the Author):

Although the authors tried to improve their manuscript "A RuBisCO-participating, novel CO₂ fixation pathway in methanogenic archaea" by several efforts including ¹³C labelling studies, reanalysing phylogenies etc., the major question i.e. what is the fate of formaldehyde is so far - in my opinion - not satisfactory answered. Without that information the proposed pathway is far from being a complete CO₂ fixation pathway. It just provide some evidence that PRK and RuBisCO are involved in biomass synthesis under autotrophic conditions. Formaldehyde, the product of the RHP pathway, has to be fixed itself into biomass otherwise the story would not make any sense. If the formaldehyde channeled into methanogenesis as suggested as one possibility by the authors CO₂ would be reduced to formaldehyde at the expense of three ATP and then would be further reduced to methane yielding roughly only 1 ATP, even under high H₂ partial pressure this will never end up in a positive energy balance. The second, more plausible, explanation would be a concerted action of the proposed RHP pathway and the reductive acetyl-CoA pathway. Then, one CO₂ is fixed in the RHP pathway

yielding one molecule of formaldehyde which is then converted via the reductive pathway, Pyr:Fd OR, PEP synthetase finally to 3-PGA. Such a pathway would result in a yield of three fixed CO₂ molecules into one molecule of 3-PGA which would keep the running. Also, the energetics would be more efficient: The fixation of three CO₂ into one 3-PGA would cost roughly 4 ATP which is in the same range as the reductive acetyl-CoA pathway, the energetically most effective CO₂ fixation pathway described so far. However, the results presented by the authors appear to argue against this possibility, and thus the manuscript still lacks an explanation how the proposed metabolism would not run out of and how the energetic balance becomes positive. The suggestion that the pentose biphosphate pathway might act as anaplerotic sequence appears also not valid since also nucleosides to be synthesised at last from CO₂ under autotrophic conditions, but an explanation how is achieved is not given.

Altogether, the authors need in my opinion to provide a reasonable explanation how formaldehyde is converted into biomass.

Our response:

Thank you for your important suggestions to improve our manuscript. We accept that our explanation of the idea concerning the RHP pathway with respect to energy and biomass not sufficiently clear in the original manuscript.

Certainly, the RHP pathway requires a relatively high energy cost, but we expect that energy for the RHP pathway can be provided from methanogenesis. Methanogenesis depends on hydrogen, independently of providing reduced carbon from the RHP cycle, and, therefore, high methane and energy production can be achieved under high hydrogen conditions, as reported previously (Conrad, 1999). Therefore, we expect that the RHP pathway can work under conditions where methanogenesis is sufficiently active to produce enough energy for the RHP pathway and other pathways essential for growth, even though the energy yield of methanogenesis is low, with 1 mol of ATP synthesized per 1 mol of methane produced. Actually, our experimental conditions included high hydrogen and CO₂

concentrations that should sustain high levels of methanogenesis. In addition, we used JCM medium 242 containing yeast extract and trypticase peptone for heterotrophic culture conditions, which was established as suitable for *M. hungatei* growth because *M. hungatei* can grow under autotrophic and heterotrophic conditions (Ferry and Wolfe, 1977). These heterotrophic conditions might also contribute to enable/facilitate the RHP pathway. The heterotrophic medium can provide nucleosides for the pentose bisphosphate pathway, which is an anaplerotic pathway to supply RuBP to the RHP pathway, as previously reported in *Thermococcus kodakaraensis*. Furthermore, CE-MS analysis showed that a large proportion of the carbon fixed by RuBisCO flowed to gluconeogenesis and glycolysis, and that a small fraction of this carbon was released as formaldehyde during RuBP regeneration, suggesting that metabolic flux of the RHP pathway was small and required relatively low energy, which can be sustained by methanogenesis. Considering the above, we expect that the RHP pathway can work – at least under active methanogenesis and/or heterotrophic conditions. The energetics of the RHP pathway in methanogenic Archaea under ecological conditions remains to be addressed further.

As suggested by reviewer #2, it may be energetically more effective that formaldehyde produced by the RHP pathway is converted to PGA via the reductive acetyl-CoA pathway and by pyruvate ferredoxin oxidoreductase, phosphoenolpyruvate synthase, phosphopyruvate hydratase, and phosphoglycerate mutase metabolic reactions. However, our ¹³C labeling studies suggested that the reductive acetyl-CoA pathway was active but the metabolic flow from acetyl-CoA to PGA was not observed. Therefore, regarding the fate of formaldehyde from the RHP pathway, we expect that formaldehyde is incorporated into acetyl-CoA in the reductive acetyl-CoA pathway and then converted to lipid via the mevalonate pathway, which is the archaeal isoprenoid biosynthetic pathway (Jain et al., 2014).

The above discussions were added to the discussion section of the revised manuscript.

Reviewer #3 (Remarks to the Author):

The revised version of the manuscript, A RuBisCO-participating, novel CO₂ fixation cycle in methanogenic Archaea, has adequately addressed and discussed the comments suggested by reviewer #3. The reviewer has only several minor/trivial points for consideration.

1. Pg2, line 1:

Two of the enzymes should read Two enzymes

Our response:

Thank you for your helpful suggestions. We corrected this error as suggested.

2. Pg3, lines 5-6: to generate RuBP from NMP for RuBisCO has been especially well characterized can be modified to to generate RuBP from nucleosides or NMP for RuBisCO has been especially well characterized

Same comment for Pg 9, line 4.

This can also be incorporated into the top of Figure 2, but I leave this to the authors' judgment.

Our response:

We modified 'NMP' to 'nucleosides or NMP' in the revised main text and Figure 2 (Figure 4 in the revised manuscript).

3. Pg4, line 9 from bottom: Can the authors insert the identity here? in spite of their low amino acid identity (xx%).

Our response:

We inserted the identity of 25% in the revised manuscript.

4. Pg 6, line 10: Archaea do not possess genes for transketolase and transaldolase needed for the oxidative pentose phosphate pathway

I think the authors are referring to the non-oxidative pentose phosphate pathway here.

Our response:

As suggested by reviewer #3, we mean the non-oxidative pentose phosphate pathway. We corrected the revised manuscript accordingly.

5. Supplementary Figure 1, Legend lines 1&2: phylogenetic

Our response:

We modified to phylogenetic in the revised manuscript.

6. Supplementary Figure 8, Legend

lines 1&2: phylogenetic

line 2: is the multiple sequence alignment intended to be shown or is this just referring to the process of tree construction?

Species abbreviations: are only those not designated in Supplementary Figure 1 written here?

Our response:

We corrected to “phylogenetic” in the revised manuscript on lines 1 and 2 of the legend to Supplementary Figure 8 (Figure 3 in the revised manuscript).

The multiple sequence alignment was just referring to the process of phylogenetic tree construction. We modified to “The phylogenetic tree was produced using CLUSTALW.” in the revised manuscript. The legend to Supplementary Figure 1 (Figure 1 in the revised manuscript) was also modified in the same way.

Species abbreviations that were not designated in Supplementary Figure 1 (Figure 1 in the revised manuscript) were only described in Supplementary Figure 8 (Figure 3 in the revised manuscript).

7. Supplementary Table 1: should read CO dehydrogenase

Our response:

We modified to “CO dehydrogenase” in the revised manuscript.

8. Table 1: should read 20.7 ± 1.7

Our response:

We understand reviewer #3's comment to mean that the decimal place should be consistent.

We therefore modified to “ 20.70 ± 1.70 ” in the revised manuscript.

9. Pg 11, Line 5, The authors refer to bootstrap analyses here. Either include values in the trees or modify the text.

Our response:

Significant bootstrap values (>75%) are shown on the nodes as black filled circles in Figure 1 and Figure 3 in the revised manuscript.

REVIEWERS' COMMENTS:

Reviewer #2 (Remarks to the Author):

My two major questions were (i) what is the fate of formaldehyde and (ii) what are the anaplerotic reactions of the pathway. Concerning the first point, the authors propose that formaldehyde might predominantly be incorporated into lipids (and not feed into methanogenesis what would mean to waste much energy). This might of course be, and therefore the explanation might be accepted, data however are not presented as far as I see.

The second point is – more importantly - explained by the heterotrophic medium used for growth of *M. hungatei* which provides nucleosides and NMP for the pentose bisphosphate pathway acting as an anaplerotic sequence for the RHP pathway. This heterotrophic life style of *M. hungatei* has not been clearly stated so far and appears conclusive as explanation.

This would however mean, that the proposed RHP pathway does not promote autotrophic growth of *M. hungatei*, this was at least not shown in the present study! In this respect the proposed pathway significantly differ from the other six CO₂ fixation pathways described so far which all allow for "real" autotrophy, i.e. growth exclusively with CO₂ as carbon source. In this light, the proposed RHP pathway appears more like e.g the PEP carboxykinase or puruvate: Fd OR which also fix CO₂ but not as their main function in the carbon metabolic network of heterotrophs (and nobody calls these reactions CO₂ fixation pathway).

All in all, the authors presented the characterization of the first archaeal PRK including the crystal structure and they propose a function of the enzyme in the carbon metabolic network of *M. hungatei*, which is undoubtable a very nice piece of work. They could, however, not convince me by the presented data and explanations/discussions that the proposed RHP pathway really represent the 7th CO₂ fixation pathway as implied in the manuscript title.

Reviewers' comments:

Reviewer #2 (Remarks to the Author):

My two major questions were (i) what is the fate of formaldehyde and (ii) what are the anaplerotic reactions of the pathway. Concerning the first point, the authors propose that formaldehyde might predominantly be incorporated into lipids (and not feed into methanogenesis what would mean to waste much energy). This might of course be, and therefore the explanation might be accepted, data however are not presented as far as I see. The second point is - more importantly - explained by the heterotrophic medium used for growth of *M. hungatei* which provides nucleosides and NMP for the pentose biphosphate pathway acting as an anaplerotic sequence for the RHP pathway. This heterotrophic life style of *M. hungatei* has not been clearly stated so far and appears conclusive as explanation.

This would however mean, that the proposed RHP pathway does not promote autotrophic growth of *M. hungatei*, this was at least not shown in the present study! In this respect the proposed pathway significantly differ from the other six CO₂ fixation pathways described so far which all allow for "real" autotrophy, i.e. growth exclusively with CO₂ as carbon source. In this light, the proposed RHP pathway appears more like e.g the PEP carboxykinase or puruvate: Fd OR which also fix CO₂ but not as their main function in the carbon metabolic network of heterotrophs (and nobody calls these reactions CO₂ fixation pathway).

All in all, the authors presented the characterization of the first archaeal PRK including the crystal structure and they propose a function of the enzyme in the carbon metabolic network of *M. hungatei*, which is undoubtable a very nice piece of work. They could, however, not convince me by the presented data and explanations/discussions that the proposed RHP pathway really represent the 7th CO₂ fixation pathway as implied in the manuscript title.

Our response:

Thank you for your important suggestions to improve our manuscript. As suggested by reviewer #2, life style of *M. hungatei* has not been clearly stated autotrophic or heterotrophic. *M. hungatei* can grow under autotrophic and heterotrophic conditions and growth rate in heterotrophic condition is much higher than that in autotrophic condition, as reported by Ferry and Wolfe. The heterotrophic medium has been established for *M. hungatei* culture and used in many experiments previously reported. Our experiment conditions also followed the established heterotrophic medium. Therefore, our data supported that the RHP pathway works at least in heterotrophic condition. However, it is unclear that the RHP pathway works in autotrophic condition in present. Further work is needed to analyze whether this pathway allows for autotrophy, that is, growth exclusively with CO₂ as carbon source. These discussions were added in abstract and discussion sections as below;

In abstract, page 2, lines 12–14,

We added the sentence, “Further work is needed to test whether the RHP pathway allows for autotrophy, that is, growth exclusively with CO₂ as carbon source.”

In discussion, bottom line in page 8 to lines 1–2 in page 9,

We added the sentence, “Whether the RHP pathway allows for autotrophy, that is, growth exclusively with CO₂ as carbon source, is unknown at present.”

In above situation, we accept that it is not proper to call the RHP pathway as the novel CO₂ fixation pathway because it is unclear that the RHP pathway

promote autotrophic growth of *M. hungatei*, as pointed out by reviewer #2. Therefore, we changed the manuscript title and expressions in the text from “CO₂ fixation pathway” to “carbon metabolism”. In addition, we removed a word, 7th CO₂ fixation pathway.